# Linoleic acid improves PIEZO2 dysfunction in a mouse model of Angelman Syndrome

Luis O. Romero [1,2,10], Rebeca Caires [1,10], A. Kaitlyn Victor[3], Juanma Ramirez[4], Francisco J. Sierra-Valdez [5], Patrick Walsh[6], Vincent Truong[6], Jungsoo Lee [1], Ugo Mayor [4,7], Lawrence T. Reiter[3,8,9], Valeria Vásquez [1] ✉ & Julio F. Cordero-Morales [1] ✉

Angelman syndrome (AS) is a neurogenetic disorder characterized by intellectual disability and atypical behaviors. AS results from loss of expression of the E3 ubiquitin-protein ligase UBE3A from the maternal allele in neurons. Individuals with AS display impaired coordination, poor balance, and gait ataxia. PIEZO2 is a mechanosensitive ion channel essential for coordination and balance. Here, we report that PIEZO2 activity is reduced in *Ube3a* deficient male and female mouse sensory neurons, a human Merkel cell carcinoma cell line and female human iPSC-derived sensory neurons with *UBE3A* knock-down, and de-identified stem cell-derived neurons from individuals with AS. We find that loss of UBE3A decreases actin filaments and reduces PIEZO2 expression and function. A linoleic acid (LA)-enriched diet increases PIEZO2 activity, mechano-excitability, and improves gait in male AS mice. Finally, LA supplementation increases PIEZO2 function in stem cell-derived neurons from individuals with AS. We propose a mechanism whereby loss of *UBE3A* expression reduces PIEZO2 function and identified a fatty acid that enhances channel activity and ameliorates AS-associated mechano-sensory deficits.

Angelman syndrome (AS) is a neurogenetic disorder characterized by cognitive, motor, and behavioral abnormalities[1]. Individuals with AS display impaired motor coordination (e.g., unable to reach objects), abnormal gait (i.e., instability while walking), sensory ataxia, scoliosis, seizures, an abnormally happy disposition, and intellectual disability[2–4]. Likewise, AS mouse models have clearly defined phenotypes resembling behavioral dysfunction, such as ataxic gait, seizures, and learning deficits, making this model useful for testing therapeutics[5–8]. The AS phenotype results from the loss of expression of an E3 ubiquitin-protein ligase (*UBE3A*) from the maternal allele[9–11].

*UBE3A* is regulated by genomic imprinting, a process that causes genes to be monoallelically expressed in neurons[12]. The cellular role of the UBE3A ubiquitin ligase is to transfer a single ubiquitin moiety from the E2 protein to a substrate protein[13]. UBE3A targets proteins for degradation, regulates their trafficking, and/or modulates their function[14–16].

The abnormal gait associated with this disorder is debilitating, as most children have difficulty walking[2]. Importantly, gait (ataxic or broad-based) is among the most common behaviors (88%) observed in children with AS[4]. *UBE3A* is imprinted in most brain neurons[1]. For

[1]Department of Physiology, College of Medicine, University of Tennessee Health Science Center, Memphis, TN 38103, USA. [2]Integrated Biomedical Sciences Graduate Program, College of Graduate Health Sciences, Memphis, TN 38163, USA. [3]Department of Neurology, College of Medicine, University of Tennessee Health Science Center, Memphis, TN 38103, USA. [4]Department of Biochemistry and Molecular Biology, Faculty of Science and Technology, UPV/EHU, Leioa, Bizkaia, Spain. [5]School of Engineering and Sciences, Tecnológico de Monterrey, Ave. Eugenio Garza Sada 2501 Sur, Monterrey 64849, Mexico. [6]Anatomic Incorporated, Minneapolis, MN 55455, USA. [7]Ikerbasque, Basque Foundation for Science, Bilbao, Bizkaia, Spain. [8]Department of Anatomy and Neurobiology, College of Medicine, University of Tennessee Health Science Center, Memphis, TN 38104, USA. [9]Department of Pediatrics, College of Medicine, University of Tennessee Health Science Center, Memphis, TN 38104, USA. [10]These authors contributed equally: Luis O. Romero, Rebeca Caires. ✉ e-mail: vvasquez@uthsc.edu; jcordero@uthsc.edu

instance, *UBE3A* exhibits maternal allele-specific expression in the Purkinje cell layer of the cerebellum and cell bodies of hippocampal CA1 – CA2 neurons in mice and humans[17–19]. Bruinsma et al. found that cerebellar function is only partly responsible for the behavioral deficits (e.g., ataxic-like gait, problems with balance) displayed by a mouse model of AS[20]. On the other hand, in the peripheral nervous system, it has been shown that proprioceptive and mechanosensitive dorsal root ganglia (DRG) neurons express the maternal *UBE3A* allele, while the paternal allele is silenced by an antisense transcript[21]. Hence, the loss of function of the maternal allele, in sensory neurons, could contribute to the phenotypes observed in AS.

Proprioception confers the ability to sense movement, tension, balance, and limb position[22]. The mechanosensitive ion channel PIEZO2 is expressed in sensory neurons innervating muscle spindles and Golgi tendon organs, where it mediates proprioception and balance[23–26]. A previous study demonstrated that *PIEZO2* deficiency in humans profoundly decreases proprioception, leading to sensory ataxia[23]. For example, a premature stop codon in *PIEZO2* causes unsteady gait and increased stride-to-stride variability in step length and force, among other deficits[23,27]. Likewise, mice lacking *Piezo2* display abnormal limb position and coordination, unstable gait, and balance deficits[24,26,28]. Moreover, *PIEZO2* is highly expressed in human Merkel cells and their afferent sensory neurons, where it mediates gentle touch and vibration[29–32]. Individuals carrying loss-of-function variants in *PIEZO2* have a selective loss of discriminative touch perception[23]. Similarly, mice deficient for *Piezo2* in the skin display reduced behavioral responses to gentle touch[29,30,32].

The notable similarities between proprioception phenotypes in individuals with AS or *PIEZO2* loss of function (LOF) mutations, as well as their associated mouse models, raise the intriguing hypothesis that mechanotransduction could be impaired in AS. However, it is unclear if the loss of maternal *Ube3a* expression in DRG neurons affects PIEZO2 activity. Here, we show that PIEZO2 function is reduced in AS and that a safflower oil diet, enriched in linoleic acid (LA), increases PIEZO2 activity, mechano-excitability, and ameliorates gait ataxia in a mouse model of AS.

## Results

### *Ube3a*$^{m-/p+}$ DRG neurons have decreased mechano-currents and -excitability

We used mice carrying a LOF *Ube3a* mutation on the C57BL/6 genetic background[5]. Heterozygous mice with a maternal deficiency (*Ube3a*$^{m-/p+}$) display AS-associated phenotypes, including lack of balance and gait ataxia. Conversely, mice with a paternal deficiency (*Ube3a*$^{m+/p-}$) do not show imbalance, unsteady gait, or sensory ataxia phenotypes[5,6,33]. Since mice lacking *Piezo2* experience severe mechanosensory and proprioceptive deficits[24], we hypothesized that DRG neurons from the *Ube3a*$^{m-/p+}$ mice would display impaired mechanical responses. Parenthetically, ~ 80% of cultured mouse DRG neurons display mechanically activated currents[34]. These mechanocurrents display various inactivation kinetics (i.e., rapidly, intermediate, and slowly inactivating currents). The rapidly inactivating currents ($\tau < 10$ ms) have been previously assigned to mouse PIEZO2[35], whereas the intermediate ($10 < \tau < 30$ ms) and slowly inactivating currents ($\tau > 30$ ms) have not yet been identified.

We measured mechanocurrents in the whole-cell patch-clamp configuration from wild-type (WT), *Ube3a*$^{m-/p+}$, and *Ube3a*$^{m+/p-}$ mouse DRG neurons. All experiments in this work were performed with male mice unless noted. All the mechanocurrents (including the rapidly inactivating currents assigned to PIEZO2) from the *Ube3a*$^{m-/p+}$ neurons were significantly reduced compared to WT or *Ube3a*$^{m+/p-}$ DRG neurons (Fig. 1a-b and Supplementary Fig. 1a-b). The displacement threshold required to elicit mechanocurrents in the *Ube3a*$^{m-/p+}$ neurons was slightly higher than WT or *Ube3a*$^{m+/p-}$ (Fig. 1c). The percentage of DRG neurons featuring PIEZO2 mechanocurrents (i.e., rapidly inactivating

currents) was similar between WT, *Ube3a*$^{m-/p+}$, and *Ube3a*$^{m+/p-}$ cultures (Supplementary Fig. 1c). Moreover, the capacitance distribution of recorded neurons was skewed towards medium to large diameter cells (i.e., mechanoreceptors[36]) and, importantly, was similar for WT, *Ube3a*$^{m-/p+}$, and *Ube3a*$^{m+/p-}$ neurons (Supplementary Fig. 1d-e). Parenthetically, DRG neurons from female *Ube3a*$^{m-/p+}$ mice also displayed reduced mechanocurrents compared to WT or *Ube3a*$^{m+/p-}$ (Supplementary Fig. 1f-g). PIEZO2 mediates most of the mechano-activated excitatory currents in mouse DRG neurons[24,31]. As neurons from *Ube3a*$^{m-/p+}$ mice display decreased mechanocurrents and an increase in displacement threshold, we sought to determine the ability of these neurons to elicit mechanically-activated action potentials. *Ube3a*$^{m-/p+}$ neurons required larger indentation steps to elicit action potentials than WT or *Ube3a*$^{m+/p-}$ ($\geq 12$ μm; Fig. 1d-e). Importantly, *Ube3a*$^{m-/p+}$ neurons have similar neuronal electrical excitability compared to WT neurons (membrane capacitance, action potential amplitudes, input resistances, and minimal current threshold required to elicit an action potential; Supplementary Fig. 1d, h-k). These results demonstrate that mechanocurrents (including those from PIEZO2) and mechano-excitability are reduced in *Ube3a*$^{m-/p+}$ DRG neurons.

### *UBE3A* knockdown alters the cytoskeleton and decreases PIEZO2 function

We asked whether knocking down the expression of *UBE3A* in a human cell line by silencing RNA (siRNA) could recapitulate our findings in DRG neurons. PIEZO2 is expressed in Merkel cells (tactile epithelial cells) and their innervating afferents, where it transduces skin indentation[29,30,32]. To support functional and biochemical experiments, we utilized the human Merkel cell carcinoma cell line (MCC13), in which PIEZO2 mediates all endogenous mechanosensitive currents[37]. MCC13 cells displayed a decrease in PIEZO2 currents after knocking down the expression of *UBE3A* or *PIEZO2*, when compared with mechanocurrents from the scrambled siRNA treatment (47% and 74%, respectively; Fig. 2a). We validated the siRNA treatment by performing RT-qPCR (Supplementary Fig. 2a-b). Quantification of mRNA levels demonstrates that knocking down *UBE3A* does not affect *PIEZO2* transcripts (Supplementary Fig. 2b). As expected, decreasing the mRNA levels of *UBE3A* reduced UBE3A protein expression (Supplementary Fig. 2c-d). Notably, knocking down the expression of *UBE3A* decreased 23% of PIEZO2 membrane levels in MCC13 cells (Fig. 2b and Supplementary Fig. 2e). For electrophysiology experiments, we patched cells expressing the transfection marker siGLO green, whereas for western blots, we extracted membrane protein from a mixed culture of transfected and untransfected cells. This could explain the difference in effect size between currents and membrane protein expression reduction (47% *vs.* 23%). Conversely, UBE3A overexpression by transient transfection of *UBE3A* in MCC13 cells showed increased PIEZO2 currents and membrane expression (Fig. 2c-d and Supplementary Fig. 2f). Taken together, in both mouse DRG neurons and human cell lines, downregulation of *UBE3A* expression decreases PIEZO2 currents.

*Ube3a* deficient mice display reduced filamentous actin (F-actin) in cultured hippocampal neurons[38]. Moreover, using proteomic profiling, we have previously shown in *D. melanogaster* that *Ube3a* homozygous mutants have less F-actin, consistent with the identification of actin targets regulated by *Ube3a*[39]. PIEZO1 and PIEZO2 channel function is modulated by cytoskeletal elements[40–43]. We previously demonstrated that disrupting actin filaments with latrunculin A decreased PIEZO2 currents in N2A cells[42]. Therefore, we tested the hypothesis that loss of *UBE3A* expression could decrease PIEZO2 currents by impairing F-actin. MCC13 cells treated with latrunculin A display reduced PIEZO2 currents compared to untreated cells (Fig. 2e). Of note, we measured a decrease in PIEZO2 membrane expression levels in MCC13 cells after latrunculin A treatment, as well as a reduction in F-actin content (Fig. 2f and Supplementary Fig. 2g-i). Importantly,

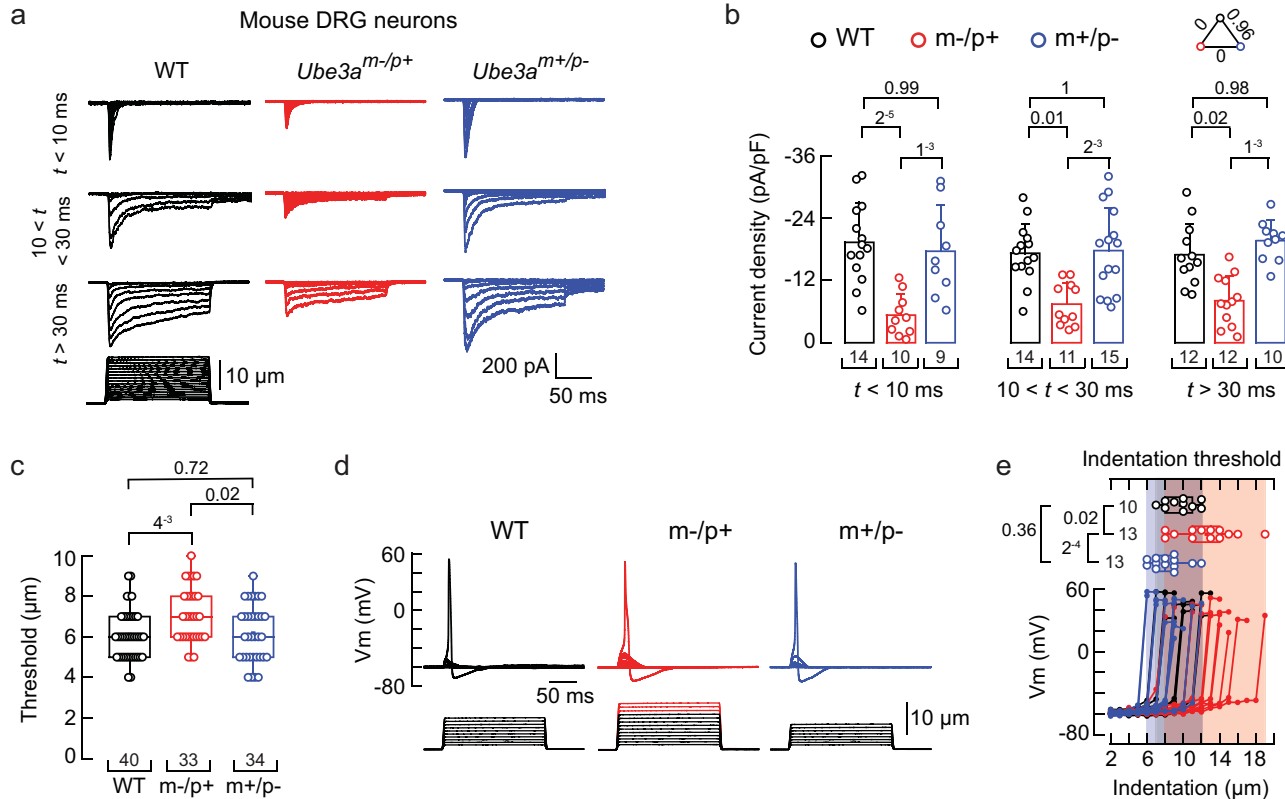

**Fig. 1 | *Ube3a^{m−/p+}* DRG neurons display reduced mechano-currents and -excitability. a** Representative whole-cell patch-clamp recording elicited by mechanical stimulation (−60 mV) of rapidly, intermediate, and slowly inactivating currents of WT, *Ube3a^{m−/p+}*, and *Ube3a^{m+/p−}* DRG neurons. **b** Current densities elicited by maximum displacement of DRG neurons classified by their time constant of inactivation. Bars are mean ± SD. Two-way ANOVA (F = 35.44, $p = 2.63^{-12}$) and Tukey multiple-comparisons test. **c** Boxplots show the displacement thresholds required to elicit mechanocurrents of DRG neurons. Boxplots show mean (square), median (bisecting line), bounds of box (75th to 25th percentiles), outlier range with 1.5 coefficient (whiskers), and minimum and maximum data points. Kruskal-Wallis (H = 9.51; p = 0.0086) and Dunn's multiple comparisons test. **d** Representative current-clamp recordings of membrane potential changes elicited by mechanical stimulation in DRG neurons. **e** Membrane potential peak vs. mechanical indentation of independent mouse DRG neurons. At the top, boxplots show the displacement threshold required to elicit an action potential in these neurons. Boxplots show mean (square), median (bisecting line), bounds of box (75th to 25th percentiles), outlier range with 1.5 coefficient (whiskers), and minimum and maximum data points. One-way ANOVA (F = 10.54; $p = 2.89^{-4}$) and Tukey multiple-comparisons test. n is denoted in each panel. Post hoc p values are denoted in the corresponding panels. Source data are provided as a Source Data file.

we also observed a reduction in actin content after knocking down the expression of *UBE3A* in MCC13 cells (Fig. 2g and Supplementary Fig. 2j). Our data support that knocking down *UBE3A* expression reduces actin filaments, leading to a decrease in PIEZO2 membrane expression and currents.

UBE3A could alter actin dynamics by regulating the content of actin-binding protein(s). Cofilin is an actin-binding protein that promotes rapid actin filament disassembly[44]. We determined an increase in cofilin content after knocking down the expression of *UBE3A* in MCC13 cells (Fig. 2h and Supplementary Fig. 2k). Moreover, overexpression of cofilin in MCC13 cells, by transient transfection, decreased PIEZO2 currents compared to control cells (Fig. 2i). Previous works have shown that cofilin can be ubiquitinated by the E3 ubiquitin ligases Cbl and AIP4[45]; however, whether UBE3A ubiquitinates cofilin is unknown. Using a stringent cell-culture based ubiquitination assay[46,47], we demonstrated that cofilin ubiquitination is increased by WT UBE3A but not by a ligase dead (LOF) version of this E3 enzyme (Fig. 2j and Supplementary Fig. 2l). Our findings support a model whereby loss of UBE3A expression increases cofilin, which severs actin filaments and decreases PIEZO2 membrane expression and currents.

### *Ube3a^{m−/p+}* DRG neurons display reduced F-actin, and promoting actin polymerization increases PIEZO2 currents

Based on our findings in MCC13 cells, we reasoned that *Ube3a^{m−/p+}* mouse DRG neurons could have a reduced F-actin content. Indeed,

cultured *Ube3a^{m−/p+}* neurons fixed and stained with Alexa Fluor 488 phalloidin (selective actin filament stain) displayed a reduced mean fluorescence intensity compared to WT (Fig. 3a). This result is further supported by the increase in the G/F actin ratio in *Ube3a^{m−/p+}* DRG neurons, as determined by western blots (Fig. 3b and Supplementary Fig. 3a). Jasplakinolide is a peptide that promotes actin polymerization and stabilizes actin filaments[48]. Notably, jasplakinolide treatment significantly increased PIEZO2 currents in *Ube3a^{m−/p+}* mouse DRG neurons (Fig. 3c). An increase in cofilin could account for the decrease in actin in the *Ube3a^{m−/p+}* mouse DRG neurons. We found elevated levels of cofilin (a target of UBE3A) in *Ube3a^{m−/p+}* mouse DRG neurons when compared to neurons from WT mice (Fig. 3d and Supplementary Fig. 3b). Importantly, knocking down the expression of *cofilin* in *Ube3a^{m−/p+}* DRG neurons increases PIEZO2 currents similar to WT levels (Fig. 3e). Taken together, our results demonstrate that *Ube3a^{m−/p+}* DRG neurons have an impaired actin cytoskeleton and treatments that stabilize actin filaments or reduce cofilin expression increase PIEZO2 function.

### Linoleic acid increases PIEZO2 currents

There are no agonists available for PIEZO2[49]. We have previously shown that PIEZO1 (a close homolog of PIEZO2) displayed slower inactivation and more mechanocurrents in plasma membranes containing high levels of linoleic acid (LA, ω-6 C18:2)[50] and Supplementary Fig. 4a. Considering the similarities between the PIEZO channels,

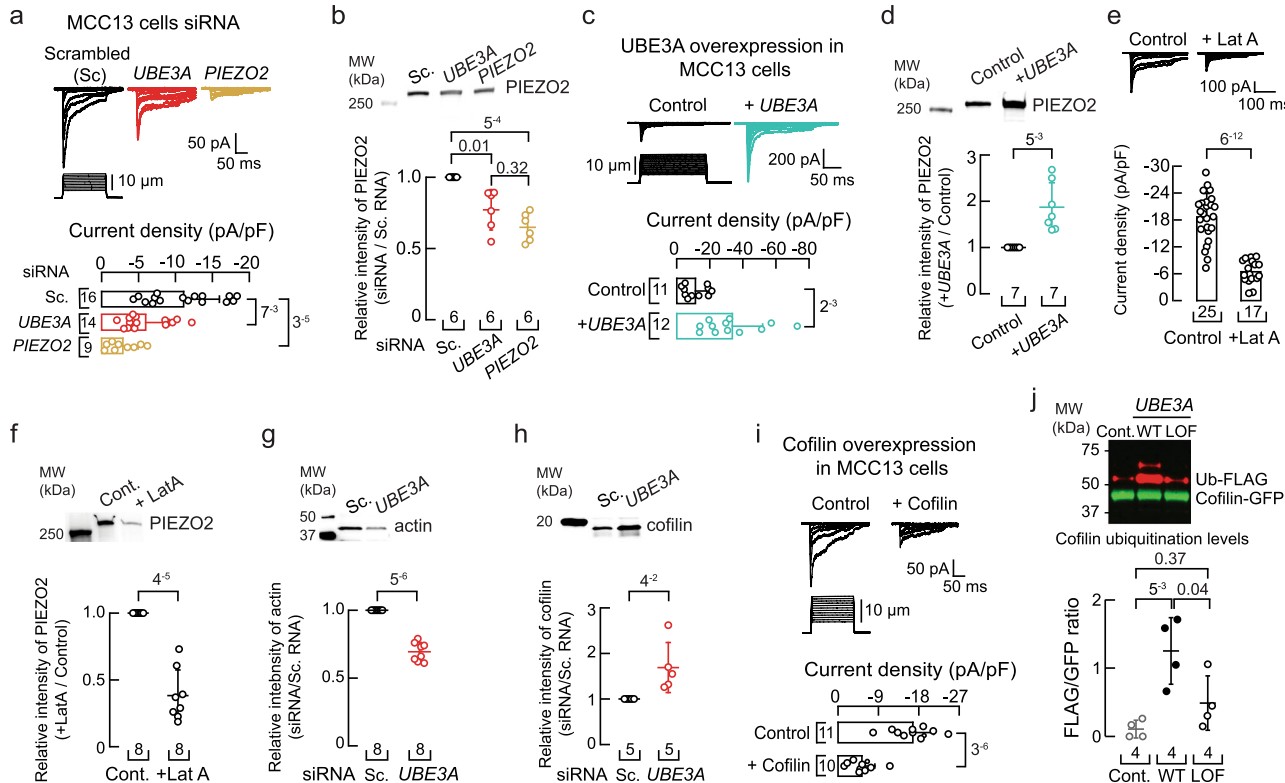

**Fig. 2 | *UBE3A* knockdown increases cofilin and decreases F-actin content and PIEZO2 function. a** Top, representative whole-cell patch-clamp recordings of currents elicited by mechanical stimulation (−60 mV) in MCC13 cells transfected with scrambled, *UBE3A*, or *PIEZO2* siRNAs. Bottom, current densities elicited by maximum displacement of siRNA-transfected cells. Bars are mean ± SD. Kruskal-Wallis (H = 18.76; $p = 8.4^{-5}$) and Dunn's multiple comparisons test. **b** Top, western blot (anti-PIEZO2) of the membrane fractions of MCC13 cells transfected as in (a). Bottom, mean/scatter-dot plot showing relative intensities of PIEZO2 protein normalized to PIEZO2 in the Sc. group. Lines are mean ± SD. Kruskal-Wallis (H = 12.78; $p = 0.0017$) and Dunn's multiple comparisons test. **c** Top, currents elicited by mechanical stimulation (−60 mV) in cells transfected with *UBE3A* plasmid. Bottom, current densities elicited by maximum displacement of *UBE3A* transfected cells. Bars are mean ± SD. Two-tailed unpaired *t*-test with Welch's correction (t = 3.9). **d** Top, western blot (anti-PIEZO2) of the membrane fractions of MCC13 cells transfected with *UBE3A* plasmid. Bottom, mean/scatter-dot plot showing relative intensities of PIEZO2 protein in *UBE3A* transfected cells normalized to PIEZO2 in the control group. Lines are mean ± SD. Two-tailed one-sample *t*-test (t = 4.4). **e** Top, currents elicited by mechanical stimulation (−60 mV) of latrunculin A (1 μM; 24 h)-treated MCC13 cells. Bottom, current densities elicited by maximum displacement. Bars are mean ± SD. Two-tailed unpaired *t*-test with Welch's correction (t = 9.9). **f** Top, western blot (anti-PIEZO2) of the membrane fractions of MCC13 cells treated

as in (e). Bottom, mean/scatter-dot plot showing relative intensities of PIEZO2 protein normalized to PIEZO2 in the control group. Lines are mean ± SD. Two-tailed one-sample *t*-test (t = −9.1). **g** Top, western blot (anti-actin) of the cytoskeletal fractions of MCC13 transfected with scrambled (Sc.) or *UBE3A* siRNAs. Bottom, mean/scatter-dot plot showing relative intensities of actin protein normalized to actin in the Sc. group. Lines are mean ± SD. Two-tailed one-sample *t*-test (t = −12.5). **h** Top, western blot (anti-cofilin) of the cytosolic fractions of MCC13 transfected as in (g). Bottom, mean/scatter-dot plot showing relative intensities of cofilin protein normalized to cofilin in the Sc. group. Lines are mean ± SD. Two-tailed one-sample *t*-test (t = 2.8). **i** Top, currents elicited by mechanical stimulation (−60 mV) in MCC13 cells transfected with *cofilin* plasmid. Bottom, current densities elicited by maximum displacement of *cofilin* transfected cells. Bars are mean ± SD. Two-tailed unpaired *t*-test with Welch's correction (t = 6.8). **j** Top, western blot of pulldown GFP-tagged cofilin from HEK293T cells transfected with a control vector (Ctrl), wild-type *UBE3A* (WT), or a catalytically inactive *UBE3A* (LOF). The ubiquitinated (Ub) fraction (red) was monitored with an anti-FLAG antibody. Bottom, mean/scatter-dot plot showing Ub-FLAG/Cofilin-GFP ratios. Lines are mean ± SD. One-way ANOVA (F = 9.86; $p = 0.0054$) and Tukey multiple-comparisons test. n is denoted in each panel. Post hoc *p*-values are denoted in the corresponding panels. Source data are provided as a Source Data file.

we tested the ability of LA to enhance PIEZO2 function. To this end, we transfected *Piezo2* variant 14 (abundant in the mouse trigeminal ganglion)[51] into N2A cells lacking *Piezo1* (*Piezo1*[−/−] N2A cells)[52] to distinguish the effect of LA on PIEZO2 gating unequivocally. PIEZO2 mechanocurrents were measured after supplementing the cell media overnight with 100 μM LA. Supplementation with LA increased PIEZO2 currents twofold (−50.09 ± 19.84 pA/pF *vs.* −106.32 ± 58.9 pA/pF, mean ± SD) (Fig. 4a-b and Supplementary Fig. 4b). Overnight incubation with LA increased (~sevenfold) the plasma membrane content of this PUFA, as determined by liquid chromatography-mass spectrometry (LC-MS; Supplementary Fig. 4c). Next, we tested fatty acids of varying acyl-chain length and unsaturations to assess the chemical and structural bases whereby LA increases PIEZO2 function. We did not observe an increase in PIEZO2 activity for stearic acid (SA; C18:0), oleic acid (OA; C18:1), ω−6 PUFAs downstream of LA [gamma linolenic acid

(γLA; C18:3), dihomo gamma-linolenic acid (DγLA; C20:3), arachidonic acid (AA; C20:4), docosatetraenoic acid (DTA; 22:4)], or ω-3 PUFAs (αLA; C18:3 and DHA; C22:6) (Fig. 4a-b and Supplementary Fig. 4d-e). Additionally, LA slowed PIEZO2 channel inactivation in *Piezo1*[−/−] N2A cells (Fig. 4a and c). These results support that LA (C18:2), but not the other fatty acids tested, enhances PIEZO2 activity.

Similar to *Piezo1*[−/−] N2A cells, we measured a significant increase in endogenous PIEZO2 currents in MCC13 cells after overnight supplementation with LA, in a dose-dependent manner (Fig. 4d-e and Supplementary Fig. 4f). We also used an alternative supplementation protocol to add lower doses of LA for several days. Supplementing MCC13 cells with 20 μM LA each day, for five days, significantly increased PIEZO2 currents (Fig. 4d-e). These concentrations are within the range of circulating fatty acids present in the blood plasma of healthy adults[53]. Our results

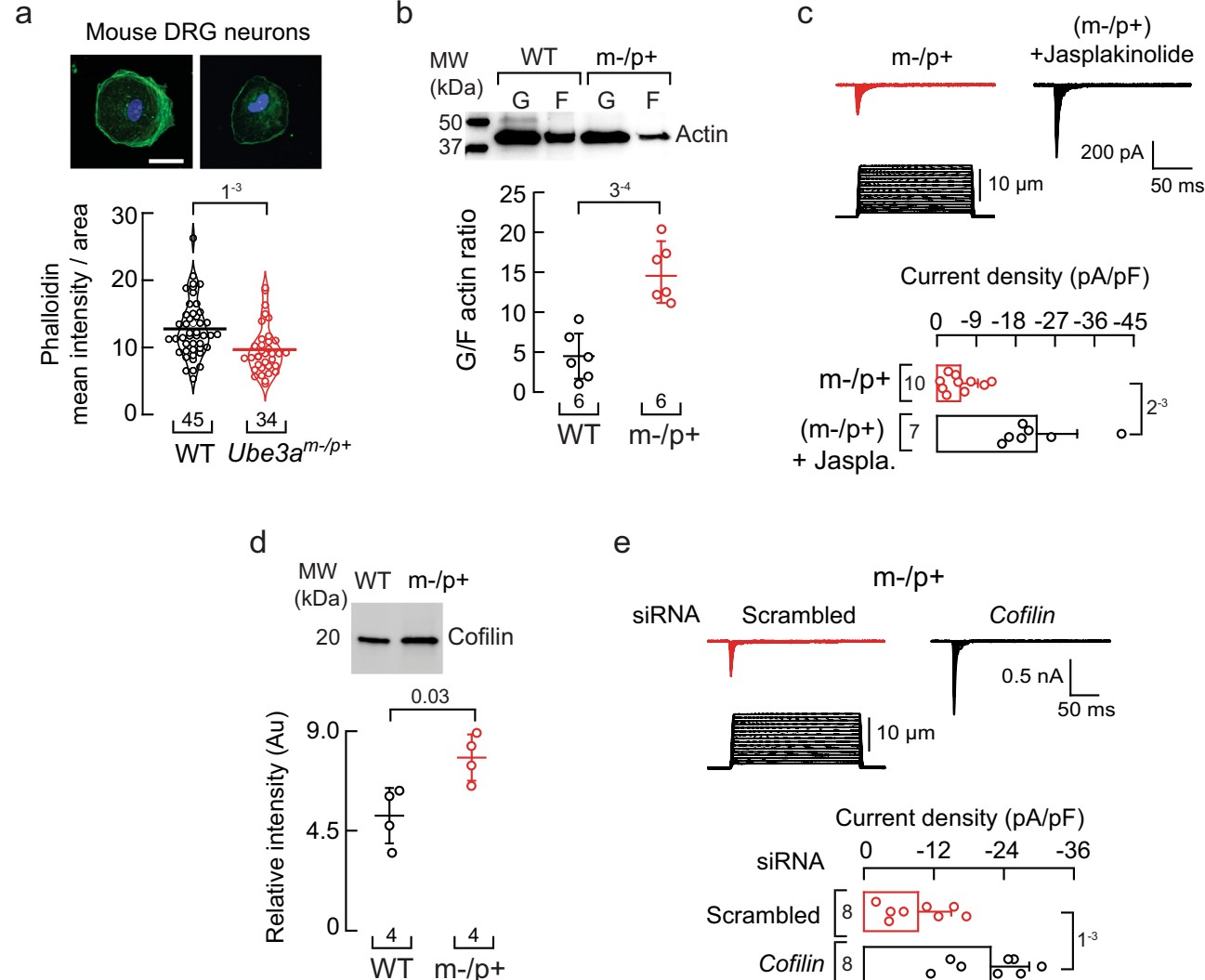

**Fig. 3 | *Ube3a^{m−/p+}* DRG neurons display reduced F-actin and jasplakinolide treatment increases PIEZO2 currents. a** Top, representative micrographs of cultured WT and *Ube3a^{m−/p+}* DRG neurons fixed and stained with phalloidin (green) and DAPI (blue). Scale bar 20 μm. Bottom, phalloidin mean intensity normalized by the neuron's area is depicted as a violin plot with the means shown as horizontal bars. Two-tailed unpaired $t$-test ($t = 3.41$). **b** Top, western blot of soluble and insoluble actin (G and F, respectively) of WT and *Ube3a^{m−/p+}* DRGs. Bottom, mean/scatter-dot plot showing G/F actin ratios. Lines are mean ± SD. Two-tailed unpaired $t$-test ($t = 5.43$). **c** Top, representative whole-cell patch-clamp recordings of PIEZO2 currents elicited by mechanical stimulation (−60 mV) of control and jasplakinolide (0.5 μM; 18 h)-treated *Ube3a^{m−/p+}* DRG neurons. Bottom, current densities elicited

by maximum displacement. Bars are mean ± SD. Two-tailed unpaired $t$-test with Welch's correction ($t = 4.68$). **d** Top, representative western blot (anti-cofilin) of the cytosolic fractions of WT and *Ube3a^{m−/p+}* DRGs. Bottom, mean/scatter-dot plot showing relative intensities of cofilin content. Lines are mean ± SD. Two-tailed Mann-Whitney test (U = 0). **e** Top, representative whole-cell patch-clamp recordings of currents elicited by mechanical stimulation (−60 mV) of *Ube3a^{m−/p+}* DRG neurons transfected with scrambled or *cofilin* siRNAs. Bottom, current densities elicited by maximum displacement of siRNA-transfected *Ube3a^{m−/p+}* DRGs. Bars are mean ± SD. Two-tailed unpaired $t$-test ($t = 4.02$). n is denoted in each panel. Post-hoc $p$-values are denoted in the corresponding panels. Source data are provided as a Source Data file.

demonstrate that the essential dietary fatty acid LA (C18:2) increases PIEZO2 activity.

## LA increases membrane structural disorder

Western blot analyses revealed that LA supplementation did not increase PIEZO2 membrane expression in MCC13 cells (Fig. 5a-b). Furthermore, perfusion of free LA (150 μM) onto *Piezo1^{−/−}* N2A cells transfected with PIEZO2 did not change channel function (Fig. 5c-d). Hence, it is possible that LA increases PIEZO2 function by modifying the membrane's mechanical properties. To determine the effect of LA on membranes, we used differential scanning calorimetry (DSC) and measured changes in the heat-capacity profiles (Cp) of synthetic membranes (1,2-dipalmitoyl-sn-glycero-3-phosphocholine, DPPC) containing various fatty acids. From the DSC thermograms, we obtained melting transition temperature (Tm) and cooperativity (κ) as

a readout of both lipid-lipid interaction and membrane organization (Fig. 5e)[54,55]. LA displayed the lowest melting temperature (i.e., less temperature required to transition from the gel to the liquid phase) compared to DPPC alone or liposomes containing SA, OA, γLA, DγLA, AA, or DTA (Fig. 5e-f). This result indicates that LA elicits the largest increase in membrane structural disorder. We also determined that liposomes containing LA display the largest cooperative unit, indicating that there are more lipids undergoing the phase transition simultaneously (Fig. 5g). The effect of LA versus the other fatty acids is made further apparent when plotting the current densities (from Fig. 4b) as a function of cooperativity (Pearson's r: 0.93; Fig. 5h).

Next, we tested the effect of LA supplementation on the bacterial mechanosensitive ion channel of large conductance (MscL), whose gating relies solely on the mechanical properties of the plasma membrane[56]. To this end, we measured pressure-activated currents

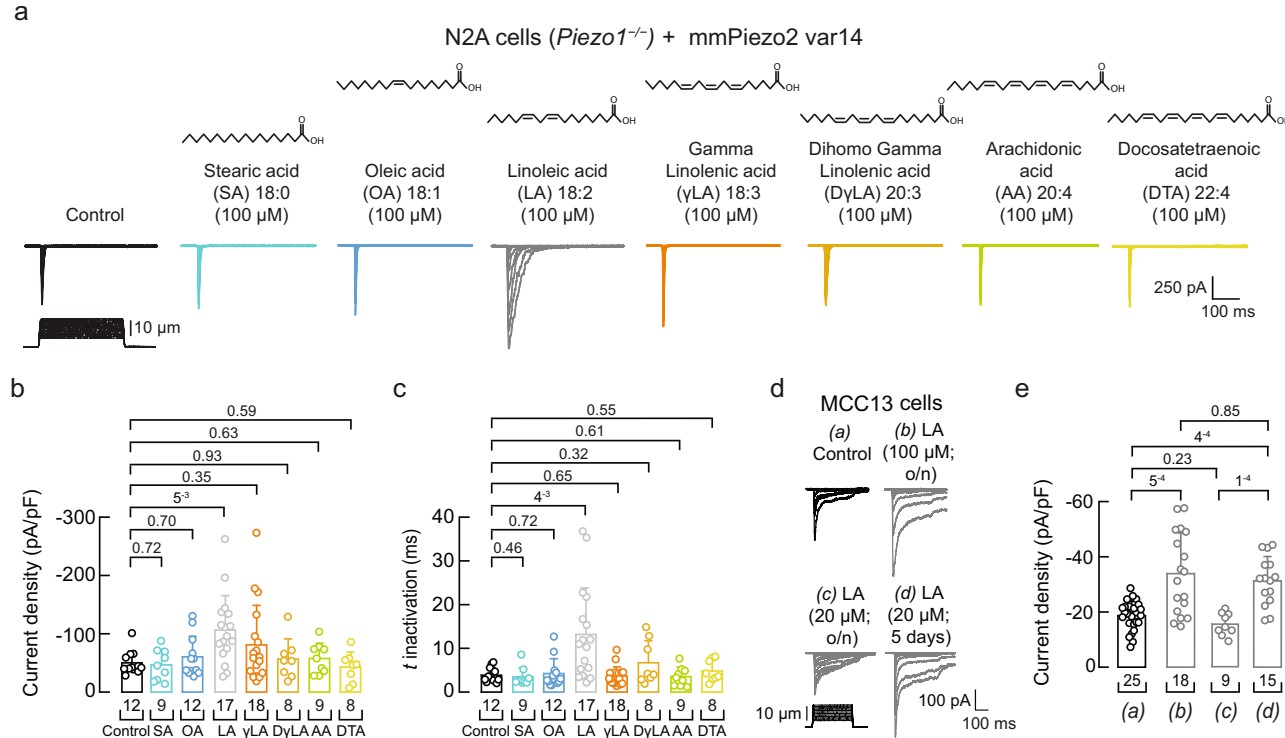

**Fig. 4 | LA increases PIEZO2 activity in *Piezo1*−/− N2A and MCC13 cells.**
**a** Representative whole-cell patch-clamp recordings of currents elicited by mechanical stimulation (−60 mV) in control, SA, OA, LA, γLA, DγLA, AA, and DTA (100 μM; 24 h)-treated *Piezo1*−/− N2A cells transfected with *Piezo2* variant 14 (var14). **b** Current densities elicited by maximum displacement of control or fatty acid-treated *Piezo1*−/− N2A cells transfected with *Piezo2* var14. Bars are mean ± SD. Kruskal-Wallis (H = 15.7; *p* = 0.028) and Dunn's multiple comparisons test. **c** Time constants of inactivation elicited by maximum displacement of control or fatty acid-treated *Piezo1*−/− N2A cells transfected with *Piezo2* var14. Bars are mean ± SD.

Kruskal-Wallis (H = 22.41; *p* = 0.0022) and Dunn's multiple comparisons test. **d** Representative whole-cell patch-clamp recordings elicited by mechanical stimulation (−60 mV) of (*a*) control, (*b*) LA (100 μM; o/n), (*c*) LA (20 μM; o/n), and (*d*) LA (20 μM each day for five days)-treated MCC13 cells. **e** Current densities elicited by maximum displacement of control and LA-treated MCC13 cells. Bars are mean ± SD. Kruskal-Wallis (H = 27.03; *p* = 5.8⁻⁶) and Dunn's multiple comparisons test. n is denoted in each panel. Post hoc *p*-values are denoted above the bars. Source data are provided as a Source Data file.

of MscL transfected in *Piezo1*−/− N2A cells, with or without LA. Similar to what has been reported for MscL reconstituted in liposomes containing LA[57], supplementing N2A cells with this fatty acid increased the function of MscL, when compared to untreated cells, as determined by the leftward shift in the pressure required to open the channel (Fig. 5i-j). Taken together, these results support the notion that the effect of LA on increasing PIEZO2 function is likely through membrane remodeling (i.e., higher membrane fluidity and cooperativity) rather than changes in protein expression or the interaction of free LA with the channel.

**Linoleic acid supplementation increases mechanocurrents in *Ube3a*m−/p+ DRG neurons**
We tested whether LA could increase mechano-responses in DRG neurons dissected and cultured from *Ube3a*m−/p+ mice. The culture media was supplemented with LA (50 μM for the first 24 h, followed by 100 μM for the second 24 h period) for two days before measuring currents. A two-day supplementation protocol improved neuronal viability and the ability to perform patch-clamp recordings. Like DRG neurons measured after 24 h (Fig. 1b), *Ube3a*m−/p+ DRG neurons displayed reduced mechanocurrents compared with WT or *Ube3a*m+/p− DRG neurons, after 48 h in culture (Supplementary Fig. 5a). Notably, LA supplementation increased all the mechanocurrents in cultured DRG neurons from WT, *Ube3a*m−/p+, and *Ube3a*m+/p− mice (Fig. 6a-d and Supplementary Fig. 5b). *Ube3a*-deficient neurons supplemented with LA displayed a decrease in the displacement threshold required to elicit mechanocurrents when compared to control, whereas no effect was measured for WT and *Ube3a*m+/p− neurons (Fig. 6e).

Parenthetically, we found that DRG neurons supplemented with LA have similar membrane capacitance, action potential amplitudes, resting potentials, input resistances, and minimal current threshold required to elicit an action potential, compared to control neurons (Supplementary Fig. 5c-h), suggesting that LA does not affect neuronal electrical excitability. Additionally, LA supplementation does not alter the function of the sensory receptors TRPV1, TRPA1, and TRPM8 (Supplementary Fig. 5i-n). Together, these results demonstrate that LA increases mechanocurrents, including those from PIEZO2, while decreasing the mechanical threshold for *Ube3a*m−/p+ DRG neurons.

**A linoleic acid-enriched diet recovers *Ube3a*m−/p+ neuronal mechano-currents and -excitability**
LA is an essential ω-6 fatty acid and a structural component of plasma membranes[58]. It is commonly found in safflower, soybean, sunflower, corn, and canola oils[59–61]. Since LA supplementation in the culture media increases mechanocurrents, we postulated that including LA in the animal's diet may rescue the mechano-deficits of *Ube3a*m−/p+ DRG neurons. To this end, we used a non-western-style diet enriched in safflower oil as a delivery method to increase LA in DRG neurons. Oil from the seeds of the safflower plant has been shown to contain over 70% LA[62]. LA can accumulate when its consumption in the diet is increased due to its limited conversion by the delta-6 desaturase enzyme[59]. We pair-fed *Ube3a*m−/p+ mice for 12 weeks with a LA-enriched diet and found that their DRG neurons had higher LA membrane content (~twofold) when compared to WT and *Ube3a*m−/p+ mice fed with a standard diet (Fig. 7a), as determined by LC-MS. On the other hand, the content of downstream

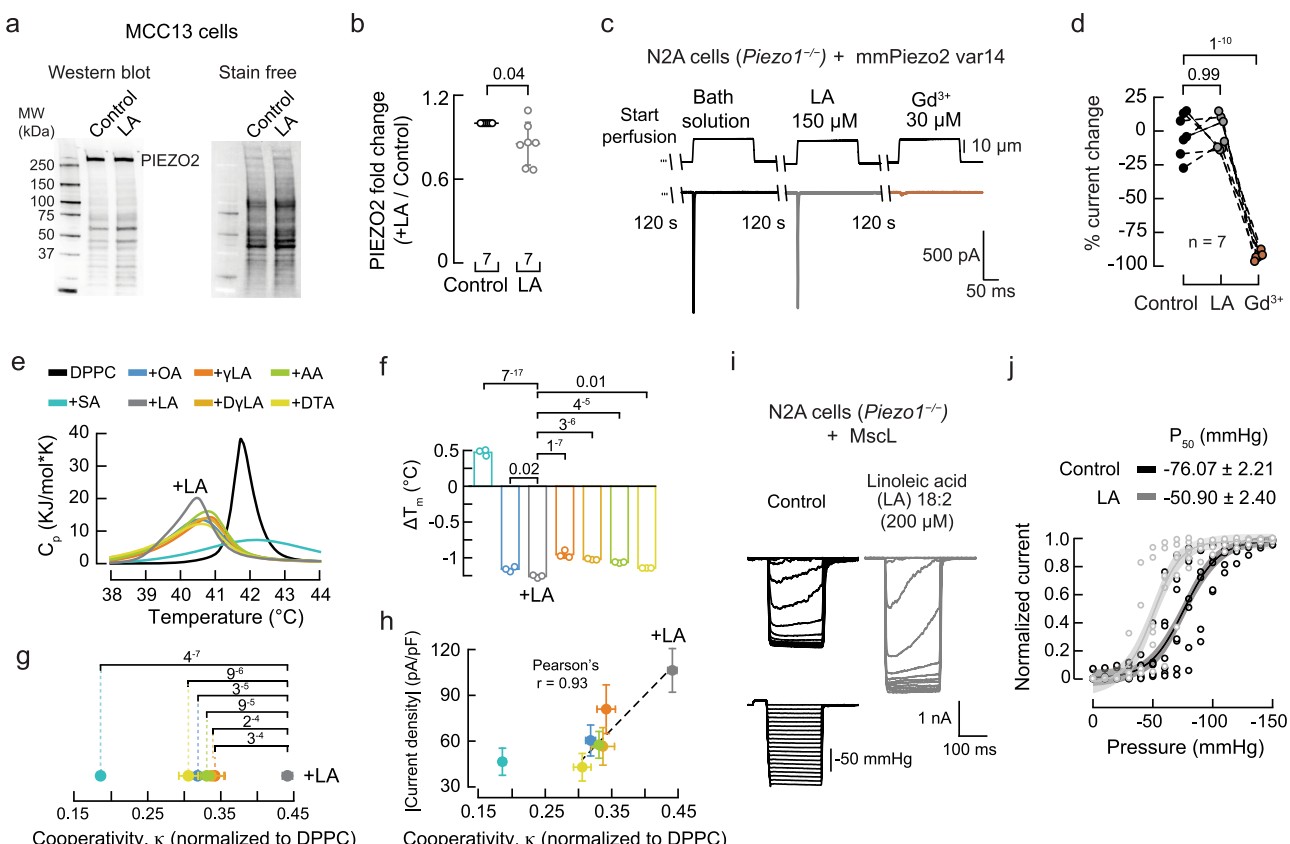

**Fig. 5 | LA does not increase PIEZO2 expression but alters the physical properties of the membranes. a** Western (anti-PIEZO2) and stain-free blots of the membrane fractions of control and LA-treated MCC13 cells. **b** Mean/scatter-dot plot showing relative intensities of PIEZO2 protein normalized to the level of PIEZO2 in the control group. Lines are mean ± SD. Two-tailed one-sample $t$-test ($t = -2.5$). **c** Representative currents elicited by 10 µm displacement of *Piezo2* var14 transfected cells after perfusing bath solution with LA and Gd³⁺. **d** Percent current change from independent cells recorded with the protocol shown in (**c**). Paired data points represent individual cells. One-sided repeated measures ANOVA ($F = 186.94$; $p = 9.51^{-6}$) and Tukey test. **e** Thermotropic characterization of the DPPC/fatty acid systems using DSC: control (Tm = 41.75 ±0.05 °C; mean ± sd), SA (42.23±0.04 °C), OA (40.59±0.03 °C), LA (40.53±0.06 °C), γLA (40.80±0.05 °C), DγLA (40.73±0.01 °C), AA (40.79±0.01 °C), and DTA (40.61±0 °C). **f** Effects of DPPC/fatty acids on melting temperatures (ΔTm) with respect to DPPC membranes. $n = 3$. Bars are mean ± SD. One-way ANOVA (F = 1,177; $p = 0$) and Bonferroni test. **g** Cooperative unit (κ) of the main transition of DPPC/fatty acid systems extracted from the thermotropic curves shown in (**e**), normalized to DPPC. Circles are mean ± SD. $n = 3$. Two-way ANOVA ($F = 45.76896$; $p = 2.09^{-8}$) and Tukey multiple-comparisons test. **h** Mean current densities of control or fatty acid-treated *Piezo1⁻/⁻* N2A cells transfected with *Piezo2* var14 vs. the cooperative unit (κ) of the main transition of DPPC/fatty acid systems. Circles are mean ± SEM. n = 3. A Pearson correlation was fitted to the unsaturated fatty acids data. **i** Inside-out recordings of currents elicited by negative pressure (at −10 mV) in LA (200 µM; 24 h)-treated cells transfected with MscL. **j** Normalized current responses to pressure changes of control ($n = 7$) and LA (200 µM; 24 h; $n = 6$)-treated cells transfected with MscL. A Boltzmann function was fitted to the data (continuous lines). The shadows indicate the 95% confidence bands. n is denoted in each panel. Post-hoc $p$-values are denoted in the corresponding panels. Source data are provided as a Source Data file.

PUFAs (γLA, DγLA, AA, and DTA) remained constant (Supplementary Fig. 6a). Remarkably, cultured DRG neurons from *Ube3a^{m−/p+}* mice fed with a LA-enriched diet displayed robust mechanocurrents and a lower displacement threshold when compared to neurons from *Ube3a^{m−/p+}* mice fed with standard or high-fat diets (Fig. 7b-d and Supplementary Fig. 6b). Although the LA-enriched diet enhanced PIEZO2- and intermediate inactivating mechanocurrents, it had no effect on those that were slowly inactivating (Fig. 7c, right panel). However, further supplementing DRG neurons from *Ube3a^{m−/p+}* mice fed with a LA-enriched diet with additional LA (during culture) increased the slowly inactivating currents (Supplementary Fig. 6c-d). Next, we tested DRG neurons of the *Ube3a^{m−/p+}* mice fed with a LA-enriched diet for their ability to elicit mechanically activated action potentials. These neurons required smaller indentation steps (≤10µm, like WT and *Ube3a^{m+/p−}* mice) to elicit action potentials compared to neurons from *Ube3a^{m−/p+}* mice on standard or high-fat diets (≥12 µm; Fig. 7e-f).

We also tested the effect of a LA-enriched diet on WT animals and found that LA increased PIEZO2 currents in neurons compared to those from animals fed with a standard diet, but did not change the displacement threshold or threshold for mechanically activated action potentials (Supplementary Fig. 6e-i). Of note, WT, *Ube3a^{m−/p+}* mice fed with a standard diet, and *Ube3a^{m−/p+}* mice fed with a LA-enriched diet had similar body weights after pair-feeding (Supplementary Fig. 7a). Furthermore, *Ube3a^{m−/p+}* mice fed with a LA-enriched diet displayed similar behavioral responses to noxious mechanical (pinprick and tail clip) and thermal stimuli (hot plate and Hargreaves), suggesting that a LA-enriched diet does not enhance nociception (Supplementary Fig. 7b-e). We also determined that feeding the LA-enriched diet to *Ube3a^{m−/p+}* mice did not alter their cytokine profile compared to those fed with the control diet, indicating that the non-western safflower oil-enriched diet does not induce inflammation (Supplementary Fig. 7f). These results demonstrate that a LA-enriched diet increases mechanocurrents, including those from PIEZO2, and is sufficient to recover the mechanical excitability of *Ube3a^{m−/p+}* DRG neurons.

**A linoleic acid-enriched diet ameliorates gait ataxia in a mouse model of AS**

We sought to determine the effect of the LA-enriched diet on the gait of *Ube3a^{m−/p+}* mice. Mouse gait measurements are consistent, commonly used to assess locomotion in several human disease models,

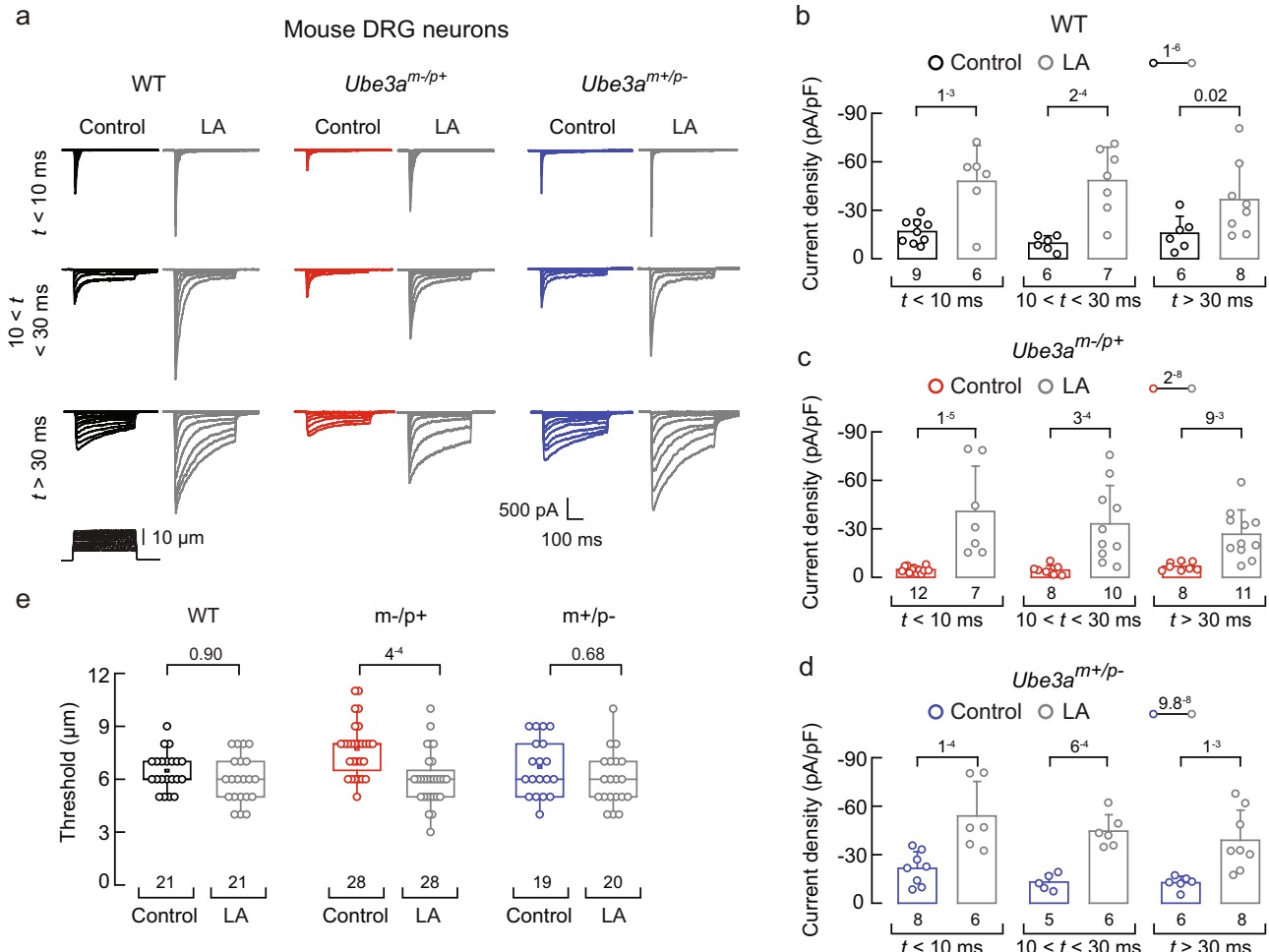

**Fig. 6 | LA increases mechanocurrents in *Ube3a*$^{m-/p+}$ DRG neurons.**
**a** Representative whole-cell patch-clamp recording elicited by mechanical stimulation (−60 mV) of rapidly, intermediate, and slowly inactivating currents of control and LA-treated WT, *Ube3a*$^{m-/p+}$, and *Ube3a*$^{m+/p-}$ DRG neurons. We used a two-day LA supplementation protocol (50 μM for 24 h and 100 μM for another 24 h). **b** Current densities elicited by maximum displacement of control and LA-treated WT DRG neurons classified by their time constant of inactivation. Bars are mean ± SD. Two-way ANOVA (F = 34.4; *p* = 1.05$^{-6}$) and Sidak–Holm multiple-comparisons test. **c** Current densities elicited by maximum displacement of control and LA-treated *Ube3a*$^{m-/p+}$ DRG neurons classified by their time constant of inactivation. Bars are mean ± SD. Two-way ANOVA (F = 43.8; *p* = 2.36$^{-8}$) and Sidak–Holm multiple-

comparisons test. **d** Current densities elicited by maximum displacement of control and LA-treated *Ube3a*$^{m+/p-}$ DRG neurons classified by their time constant of inactivation. Bars are mean ± SD. Two-way ANOVA (F = 45.9; *p* = 9.84$^{-8}$) and Sidak–Holm multiple-comparisons test. **e** Boxplots show the displacement thresholds required to elicit mechanocurrents in control and LA-treated DRG neurons. Boxplots show mean (square), median (bisecting line), bounds of box (75th to 25th percentiles), outlier range with 1.5 coefficient (whiskers), and minimum and maximum data points. Two-way ANOVA (F = 14.2; *p* = 2.59$^{-4}$) and Tukey multiple-comparisons test. n is denoted in each panel. Post hoc *p*-values are denoted above the boxes and bars. Source data are provided as a Source Data file.

and a translational metric for therapeutic development in AS[8,63,64]. We performed a gait analysis for the quantitative assessment of footfalls and locomotion using the CatWalk system (Noldus XT) for WT, *Ube3a*$^{m-/p+}$, and *Ube3a*$^{m+/p-}$ mice after feeding them with standard, LA-enriched, or high-fat diets for 12 weeks (Fig. 8a-c). *Ube3a*$^{m-/p+}$ mice fed with a standard diet displayed an increased stride length for their front and hind paws compared to the WT or *Ube3a*$^{m+/p-}$ mice fed with the same diet (Fig. 8d). We also observed a decrease in the number of steps taken during the assay for the *Ube3a*$^{m-/p+}$ mice fed with the standard diet (Fig. 8e). Petkova et al. reported a difference of ~13% between the stride length of F1 controls and *Ube3a*$^{m-/p+}$ littermate mice, which is within the range of the difference we observed of ~12–15% in our behavioral work using non-littermate mice[8]. Notably, we found that feeding *Ube3a*$^{m-/p+}$ mice with a LA-enriched diet restored the stride length and the number of steps to WT levels (Fig. 8f-g). On the other hand, feeding WT and *Ube3a*$^{m+/p-}$ mice with a LA-enriched diet did not alter their stride length or the number of steps (Supplementary Fig. 8a-d). The stride length and number of steps from *Ube3a*$^{m-/p+}$ mice fed

with a high-fat diet resemble those of the *Ube3a*$^{m-/p+}$ mice fed with a standard diet, supporting that LA, rather than the high caloric intake, ameliorates gait ataxia in *Ube3a*$^{m-/p+}$ mice. AS mouse models display impaired rotarod performance[7,10,65,66]. A LA-enriched diet did not improve *Ube3a*$^{m-/p+}$ mouse rotarod performance (Supplementary Fig. 8e), suggesting that cerebellar defects, not peripheral mechanosensory deficiencies, regulate rotarod performance in AS mice. This idea is in alignment with recent work showing that cerebellothalamic tracts mediate motor coordination but not gait[67]. Taken together, these results suggest that a LA-enriched diet can ameliorate gait ataxia (stride length and the number of steps) in a mouse model of AS.

**Knocking down *UBE3A* expression decreases PIEZO2 currents in human iPSC-derived sensory neurons**
To extend our findings to human models, we reprogrammed human-induced pluripotent stem cells (iPSCs) into sensory neurons. The human iPSC-derived neurons were manufactured using the sensory neuron differentiation kit Senso-DM (Anatomic Incorporated, Cat#

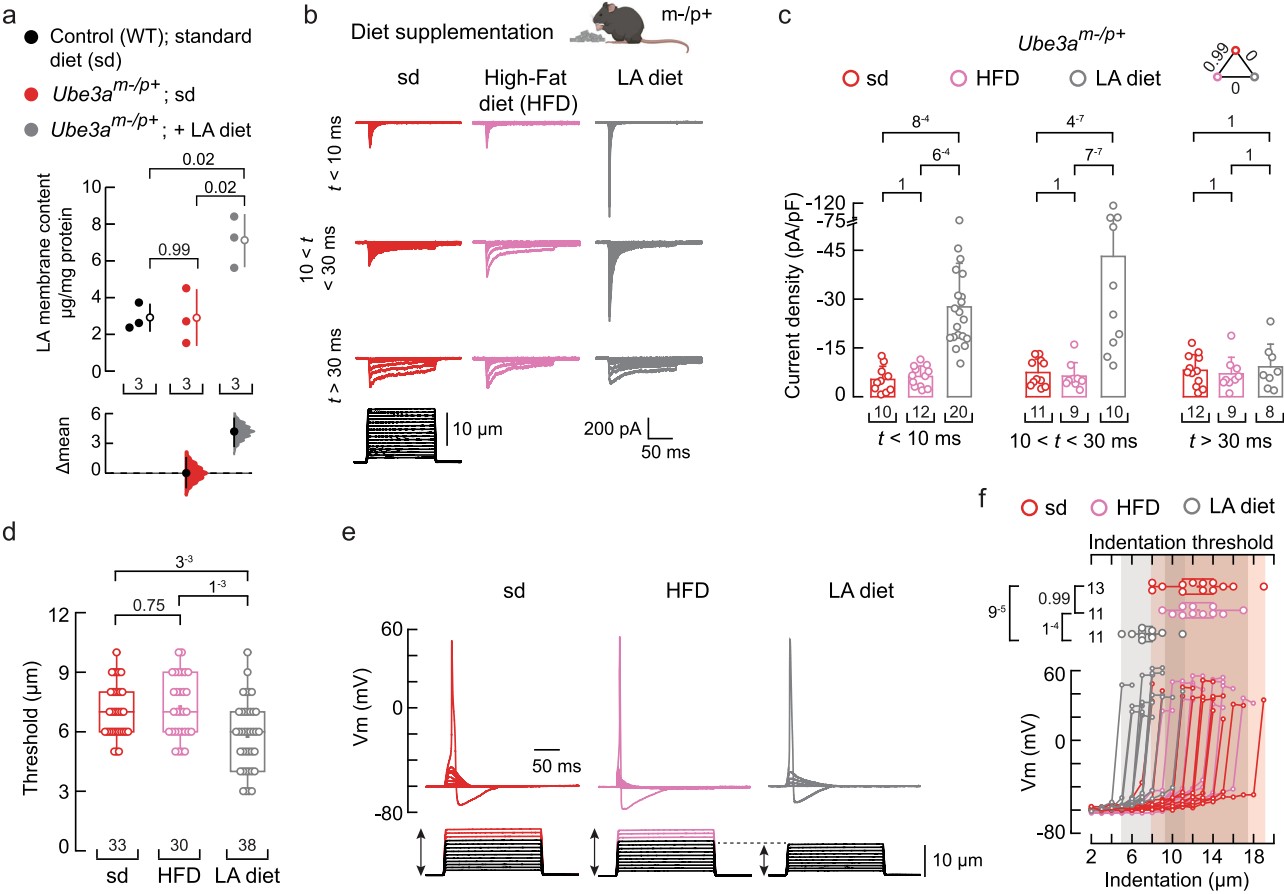

**Fig. 7 | A LA-enriched diet increases *Ube3a^(m−/p+)* mouse mechano-currents and -excitability. a** Cumming estimation plot showing the mean differences in the LA membrane content of DRG neurons from WT, *Ube3a^(m−/p+)*, and *Ube3a^(m−/p+)* mice fed with a LA-enriched diet, as determined by LC-MS. The raw data are plotted on the upper axes; each mean difference is plotted on the lower axes as a bootstrap sampling distribution. The mean differences are depicted as open circles; 95% confidence intervals are indicated by the ends of the vertical error bars. One-way ANOVA (F = 11.09; p = 0.0096) and Tukey multiple-comparisons test. **b** Representative whole-cell patch-clamp recordings elicited by mechanical stimulation (−60 mV) of rapidly, intermediate, and slowly-inactivating DRG neuron currents from *Ube3a^(m−/p+)* mice fed with a standard (sd), high-fat (HFD), or LA-enriched (LA diet) diet. Top, mouse cartoon was created with BioRender.com. **c** Current densities elicited by maximum displacement of DRG neurons from *Ube3a^(m−/p+)* mice fed with sd, HFD, or LA-enriched diets classified by their time constant of inactivation. Bars are mean ± SD. Two-way ANOVA (F = 25.36; p = 1.69^(−9))

and Tukey multiple-comparisons test. **d** Boxplots show the displacement thresholds required to elicit DRG mechanocurrents from *Ube3a^(m−/p+)* mice fed with sd, HFD, or LA-enriched diet. Kruskal-Wallis (H = 13.53; p = 0.0012) and Dunn's multiple comparisons test. **e** Representative current-clamp recordings of membrane potential changes elicited by mechanical stimulation from DRG neurons of *Ube3a^(m−/p+)* mice fed with sd, HFD, or LA-enriched diet. **f** Membrane potential peak *vs.* mechanical indentation of independent DRG neurons from *Ube3a^(m−/p+)* mice fed with sd, HFD, or LA-enriched diet. At the top, boxplots show the displacement threshold required to elicit an action potential in these neurons. One-way ANOVA (F = 14.86; p = 2.72^(−5)) and Tukey multiple-comparisons test. Boxplots show mean (square), median (bisecting line), bounds of box (75th to 25th percentiles), outlier range with 1.5 coefficient (whiskers), and minimum and maximum data points. n is denoted in each panel. Post-hoc p-values are denoted in the corresponding panels. Source data are provided as a Source Data file.

---

7007), a previously described proprietary protocol based on primal ectoderm technology[68]. We found that these in vitro-derived sensory neurons displayed mechanocurrents characteristic of PIEZO2, such as voltage-dependent inactivation and non-selective cation currents (Supplementary Fig. 9a-c). Similar to MCC13 cells, we measured a decrease in PIEZO2 currents after knocking down the expression of *UBE3A* or *PIEZO2* in these in vitro-derived sensory neurons as compared with mechanocurrents from the scrambled siRNA treatment (Fig. 9a-b). Overnight incubation of human iPSC-derived sensory neurons with LA increased endogenous PIEZO2 currents while decreasing the displacement threshold required to elicit mechanocurrents (Fig. 9c-e). Like *Piezo2* transfected in *Piezo1^(−/−)* N2A cells, LA supplementation elicited a fivefold increase in the time constant of inactivation (Supplementary Fig. 9d). Additionally, human iPSC-derived sensory neurons supplemented with LA required smaller indentation steps (≤ 6 μm) to elicit mechanically activated action potentials than untreated sensory neurons (Fig. 9f). These results

demonstrate that *UBE3A* expression modulates the function of PIEZO2, while LA increases PIEZO2 activity and enhances mechanical excitability in human iPSC-derived sensory neurons.

## Stem cell-derived neurons from individuals with AS display reduced mechanocurrents

Dental pulp stem cells (DPSC) can be differentiated into neuronal cell types and are commonly used to study neurogenetic disorders[69–72]. We have previously demonstrated that DPSC-derived neurons from individuals with AS show both molecular and cellular features characteristic of AS[72–74]. Here, we found that DPSC-derived neurons from multiple individuals with AS displayed reduced mechanocurrents, including those characteristic of PIEZO2 (Fig. 9g-h and Supplementary Fig. 9e). As expected, LA supplementation increased all mechanocurrents of DPSC-derived neurons from individuals with AS (Fig. 9g-h and Supplementary Fig. 9e). Our findings support that PIEZO2 function is decreased in neurons of mice and humans with AS and that LA

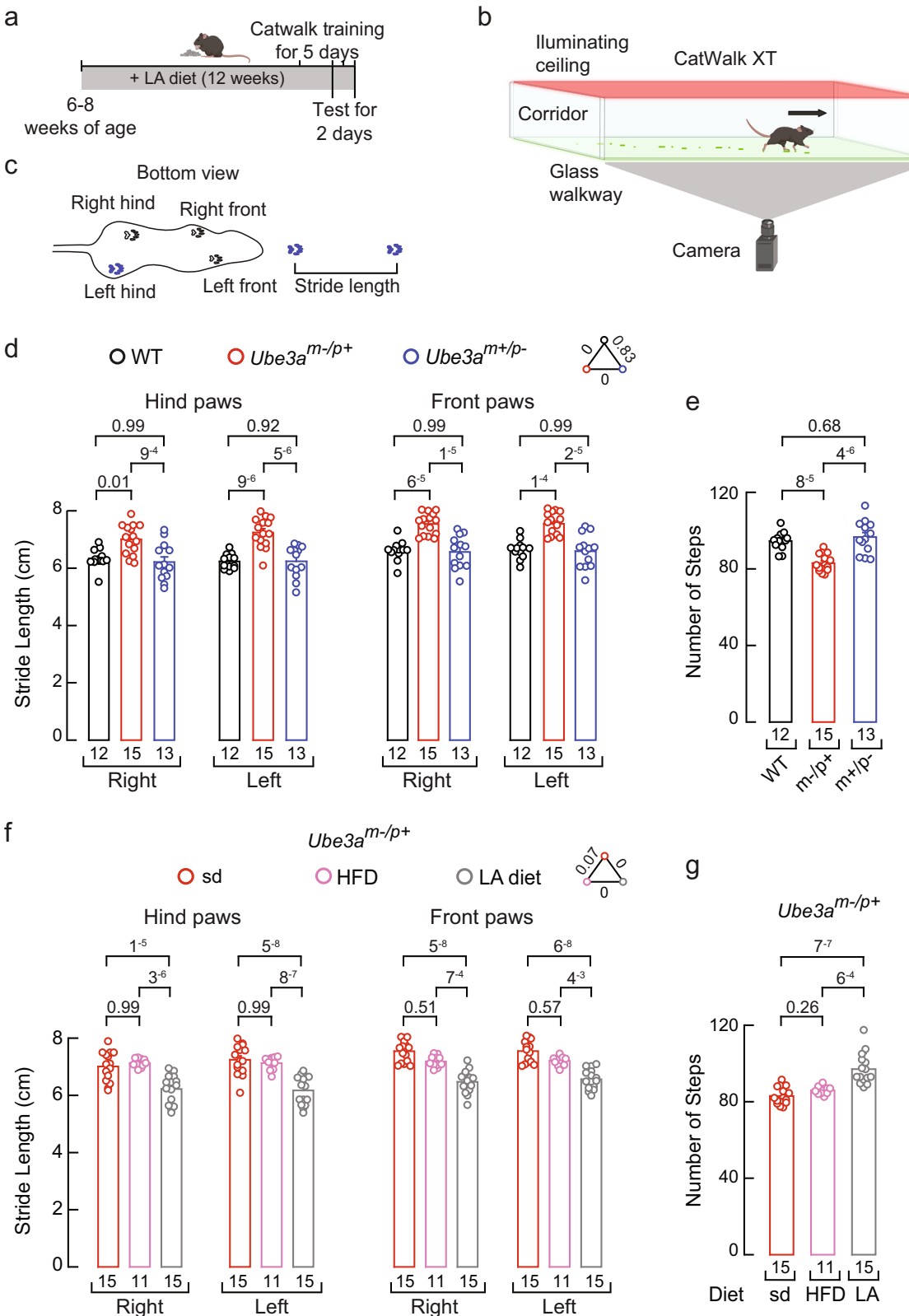

enrichment enhances PIEZO2 function, increases neurons mechanical excitability, and ameliorates gait deficits in a mouse model of AS.

## Discussion

Individuals with AS have walking and balancing difficulties, among other neurogenic-related symptoms[1]. PIEZO2 is essential for coordination and balance in mice and humans[23,24]; however, whether PIEZO2 function is altered in AS was entirely unknown. In this work, we provided several lines of evidence demonstrating that PIEZO2 currents are reduced in *Ube3a*-deficient mouse DRG neurons and stem cell-derived neurons from individuals with AS. Our results demonstrate that PIEZO2 is a therapeutic target for AS and provide proof of concept that increasing LA in the diet enhances channel activity and ameliorates gait ataxia in a mouse model of AS.

**Fig. 8 | A LA-enriched diet ameliorates gait ataxia in the _Ube3a$^{m-/p+}$_ mouse.**
**a** Timeline depicting the experimental design for gait analyses. Top, mouse cartoon was created with BioRender.com. **b** Cartoon describing gait behavioral assay was created with BioRender.com. **c** Mouse silhouette depicting the distance defined as stride length. **d** Stride length from WT, _Ube3a$^{m-/p+}$_, and _Ube3a$^{m+/p-}$_ mice fed with a standard diet. Bars are mean ± SEM. Two-way ANOVA (F = 70.2; $p = 0$) and Tukey multiple-comparisons test. **e** Number of steps from WT, _Ube3a$^{m-/p+}$_, and _Ube3a$^{m+/p-}$_ mice fed with a standard diet. Bars are mean ± SEM. One-way ANOVA (F = 19.75;

$p = 1.46^{-6}$) and Tukey multiple-comparisons test. **f** Stride length from _Ube3a$^{m-/p+}$_ mice fed with a standard, high-fat diet, or LA-enriched diet. Bars are mean ± SEM. Two-way ANOVA (F = 100.9; $p = 0$) and Tukey multiple-comparisons test. **g** Number of steps from _Ube3a$^{m-/p+}$_ mice fed with a standard, high-fat diet, or LA-enriched diet. Bars are mean ± SEM. Kruskal-Wallis (H = 26.25; $p = 1.9^{-6}$) and Dunn's multiple comparisons test. n is denoted in each panel. Post-hoc $p$-values are denoted in the corresponding panels. Source data are provided as a Source Data file.

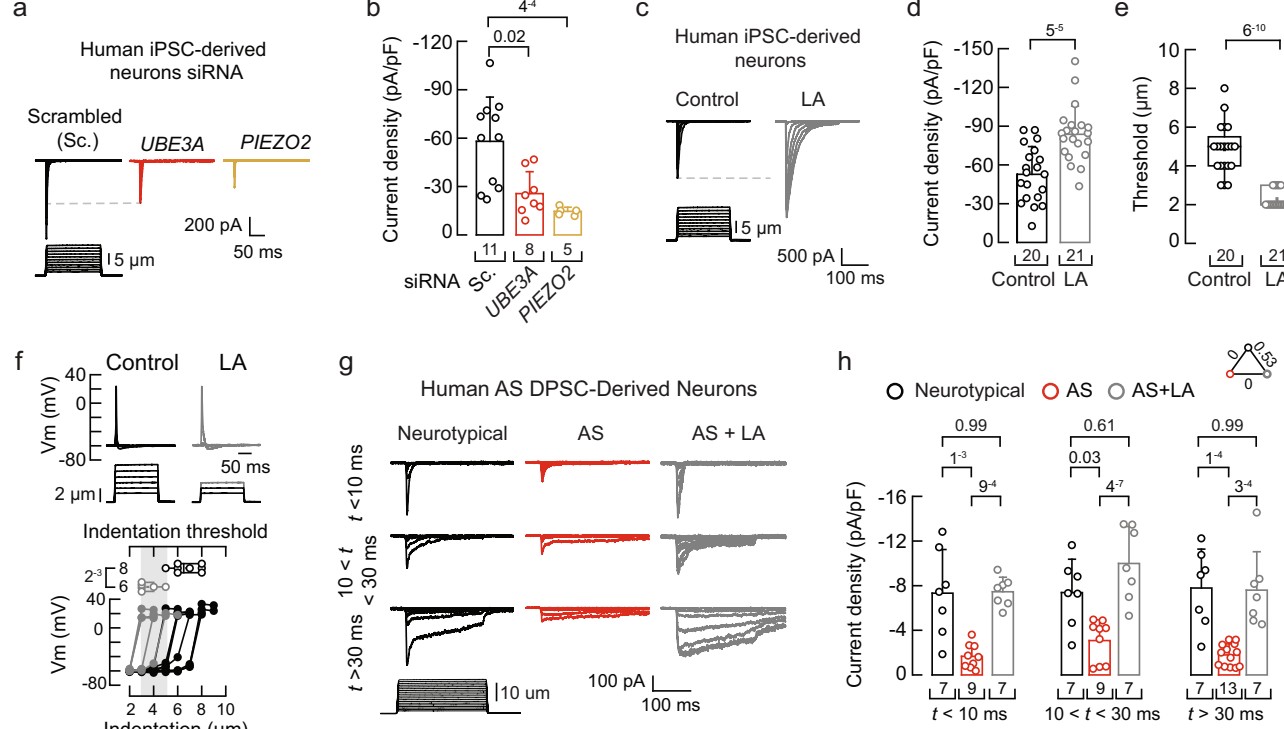

**Fig. 9 | Loss of _UBE3A_ expression decreases PIEZO2 function in human-derived neurons. a** Representative whole-cell patch-clamp recordings of currents elicited by mechanical stimulation (−60 mV) in human iPSC-derived sensory neurons transfected with scrambled, _UBE3A_, or _PIEZO2_ siRNAs. **b** Current densities elicited by maximum displacement of siRNA-transfected iPSC-derived neurons. Bars are mean ± SD. Kruskal-Wallis (H = 13.9; $p = 9.6^{-4}$) and Dunn's multiple comparisons test. **c** Representative whole-cell patch-clamp recordings of currents elicited by mechanical stimulation (−60 mV) of control and LA-treated iPSC-derived sensory neurons (two-day LA supplementation; 50 μM for 24 h and 100 μM for another 24 h). **d** Current densities elicited by maximum displacement of control and LA-treated iPSC-derived neurons. Bars are mean ± SD. Two-tailed unpaired $t$-test ($t = 4.57$). **e** Boxplots show the displacement thresholds required to elicit mechanocurrents in of control and LA-treated iPSC-derived neurons. Two-tailed Mann-Whitney test (U = 9). **f** Top, current-clamp recordings of membrane potential changes elicited by indentation of control and LA-treated iPSC-derived neurons. Bottom, membrane potential peak _vs._ mechanical indentation of independent iPSC-

derived neurons. At the top, boxplots show the displacement threshold required to elicit an action potential in these neurons. Two-tailed Mann-Whitney test (U = 0.5). **g** Representative whole-cell patch-clamp recordings elicited by mechanical stimulation (−60 mV) of rapidly, intermediate, and slowly inactivating currents of neurotypical and AS DPSC-derived neurons, with or without LA supplementation (two-day LA supplementation; 50 μM for 24 h and 100 μM for another 24 h). **h** Current densities elicited by maximum displacement from neurotypical and AS DPSC-derived neurons, ± LA supplementation, classified by their time constant of inactivation. Bars are mean ± SD. Two-way ANOVA (F = 45.23; $p = 5.69^{-13}$) and Tukey multiple-comparisons test. Neurotypical (three individuals), AS (five individuals), and AS + LA (three individuals). Boxplots show mean (square), median (bisecting line), bounds of box (75th to 25th percentiles), outlier range with 1.5 coefficient (whiskers), and minimum and maximum data points. n is denoted in each panel. Post-hoc $p$-values are denoted in the corresponding panels. Source data are provided as a Source Data file.

Previous works demonstrated that F-actin is reduced in the brain of _Ube3a_-deficient animals[38,39]. Actin is also well known to regulate membrane protein trafficking[75]. Notably, several studies indicate that an intact cytoskeleton is required for normal PIEZO2 function[35,43,52,76,77], including our previous work in which we demonstrated that PIEZO2 mechanosensitivity relies on the synergy between the mechanics of the plasma membrane and F-actin[42]. We propose a mechanism whereby loss of _UBE3A_ expression leads to an increase in cofilin, which severs actin filaments, and in turn, decreases PIEZO2 membrane expression and function in the context of AS (Fig. 10). F-actin disruption might decrease PIEZO2 activity by impairing force transmission

from the plasma membrane to the channel. Based on our data, we envision that diminished mechanical excitability, caused by reduced PIEZO2 activity, could impair somatosensory input from the limbs, resulting in the gait ataxia observed in AS. Treatments (natural and/or pharmaceutical) that increase PIEZO2 currents could bypass the deficits associated with the intracellular loss of _UBE3A_ expression and F-actin deficiency.

With the recent discovery that gain- and loss-of-function mutations in _PIEZO2_ have a detrimental effect on human skeletal development, balance, and proprioception[23,78–80], it has become clear that agonists/antagonists or molecules that modulate PIEZO2 function

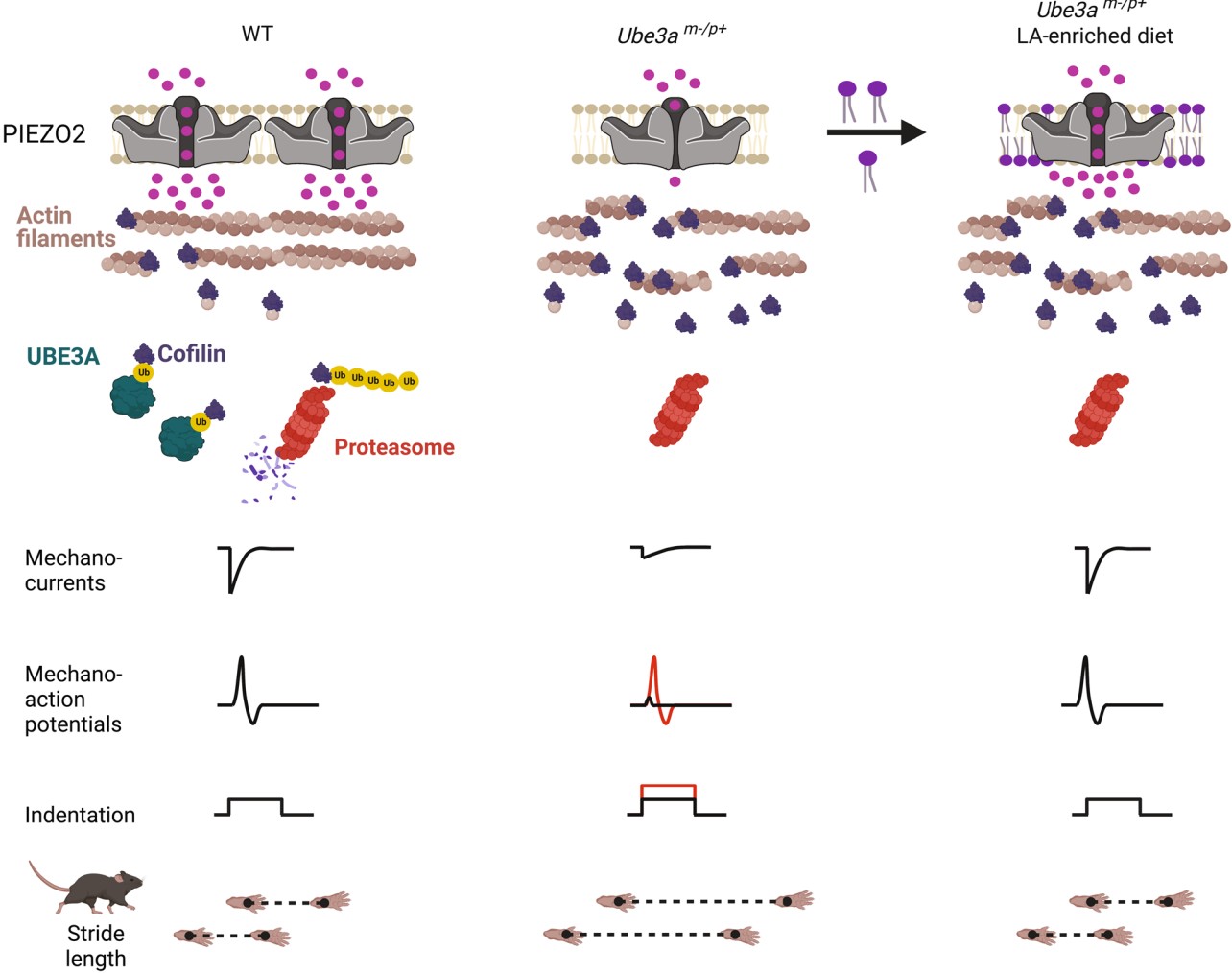

**Fig. 10 | Proposed mechanism whereby lack of UBE3A decreases mechanosensation in AS.** Loss of *Ube3a* expression increases cofilin leading to a decrease in actin filaments. Reduced actin filaments decrease PIEZO2 function as well as neuron mechano-excitability. LA increases PIEZO2 function and ameliorates gait deficits. Created with BioRender.com.

could have therapeutic effects for a wide range of pathologies. For instance, a diet enriched in eicosapentaenoic acid (EPA) decreases PIEZO2 function (by increasing channel inactivation) and counteracts the detrimental effect of a PIEZO2 gain-of-function mutation causing distal arthrogryposis in mice[81]. Currently, there are no available agonists for PIEZO2. Molecules that enhance PIEZO2 function could bypass the mechano-sensory deficits associated with the loss of *UBE3A* expression. Importantly, we have identified a fatty acid (LA) that increases PIEZO2 mechanocurrents. The ω-6 fatty acid LA, a structural component of membrane phospholipids, is an essential PUFA enriched in safflower, soybean, sunflower, corn, and canola oils[59–61]. In this work, we used a nonwestern-style diet enriched with 59% of fat-derived calories from safflower oil as a delivery method to maximize the LA content. Parenthetically, a ketogenic dietary (>60% of its calories from fat) intervention has been shown to reduce seizures in individuals with AS[82–85]. LA can accumulate when its consumption in the diet is increased due to its limited conversion by the delta-6 desaturase enzyme[59]. For instance, a systematic review of human trials demonstrated that increasing dietary LA consumption significantly increases tissue levels of LA, but not AA, in adults[59]. These trials also support that even though humans already consume LA, it can be effectively increased in tissues when further supplemented in the diet. Our data corroborate this notion since mice fed with a non-western style safflower-enriched diet resulted in an increase in LA membrane content in DRGs, with no significant changes in downstream ω-6 PUFAs (e.g., AA).

We observed that the slowly adapting mechanocurrents increase when LA is supplemented in the media, but not after feeding the *Ube3a*^m-/p+ mice with a LA-enriched diet. Since we determined that LA accumulates in membranes sevenfold when supplemented in culture media versus twofold when fed directly to mice, it is tempting to speculate that the slowly adapting mechanocurrents might require more LA accumulation than is achieved in our 12-week diet protocol. Indeed, we observed an increase in the slowly adapting mechanocurrents of DRG neurons from *Ube3a*^m-/p+ mice fed with a LA-enriched diet after supplementing them with LA in the culture media. Although dietary PUFAs share structural similarities, we previously demonstrated that EPA, but not AA, enhances TRPV4 channel activity; both PUFAs have 20 carbons, but EPA contains five unsaturations instead of four[86,87]. Here, we also revealed a degree of specificity since PIEZO2 currents were increased by LA (ω-6 C18:2) but not by SA (C18:0), OA (C18:1), PUFAs downstream of LA, αLA (ω-3 C18:3), or DHA (C22:6). These results highlight that PUFAs add a level of regulation and that subtle changes in their structure (e.g., unsaturations) differentially influence channel function.

Mechanosensitive ion channels, like PIEZO1 and PIEZO2, open by directly sensing the physical state of the plasma membrane[88]. Hence, modifying the plasma membrane by changing rigidity[42,49], local

curvature[89], organization[50], lateral tension[56], and/or hydrophobic mismatch[90], for instance, influences mechanosensitive ion channel function. Previous studies have shown that plasma membranes enriched in PUFAs display decreased membrane rigidity and enhanced function of the mechanoreceptor and photo-transduction channel complexes in *C. elegans*[91] and *D. melanogaster*[92] in vivo. Our results in synthetic DPPC liposomes agree with previous works in which LA decreased the melting transition temperature of lipid membranes[93,94]. Since the DSC results demonstrated that LA increases membrane fluidity and cooperativity in synthetic lipids, we propose that LA could exert similar effects on cell membranes and, in turn, increase PIEZO2 function by reducing its activation threshold, as determined for MscL in liposomes[57] and in this work in cells. The lack of effect on PIEZO2 mechano-responses when perfusing free LA, together with increased MscL function (a channel whose gating is solely determined by the mechanical properties of the membrane[56]) after LA supplementation, supports that the effect of LA on PIEZO2 function is likely through plasma membrane remodeling rather than direct interaction. We envision that membranes enriched in LA might favor the open state of PIEZO2 by facilitating force transmission from the blades (i.e., transmembrane domains) to the pore. Our results do not rule out specific interactions between LA (when part of the membrane) with PIEZO2. However, due to the size of PIEZO2 (~310 kDa), classical mutagenesis experiments used to pinpoint interaction sites are impractical. Interestingly, recent cryo-EM structures of PIEZO1 reconstituted in liposomes showed various bound phospholipids in the blade of the channel in the curved conformation (i.e., closed-like), but not in the flattened (i.e., open-like) one[95]. Future structural studies of PIEZO2 in the presence or absence of LA could help identify putative interaction sites that could be functionally tested.

It has been suggested that the behavioral deficits (e.g., ataxic-like gait, problems with balance) displayed by a mouse model of AS are not solely of cerebellar origin[20]. The recent discovery that *PIEZO2* deficiency in humans results in an unsteady gait and increased stride-to-stride variability and step length[23,27], together with our finding that *Ube3a*[m−/p+] DRG neurons display decreased mechanocurrents and -excitability, suggests that a decrease in PIEZO2 function could contribute to the gait ataxia experienced in AS. In this work, we focused on determining the mechanism whereby UBE3A regulates mechanosensation in AS. Even though we provided evidence that loss of *UBE3A* expression decreases PIEZO2 membrane expression and activity, we do not have direct evidence demonstrating that the reduction of PIEZO2 function is the main cause of the gait phenotype in AS. More experimental insight will be needed to determine the full implications of the reduction of PIEZO2 on gait in the context of AS. Although we found that a LA-enriched diet increases PIEZO2 activity and mechano-excitability, as well as ameliorates gait deficits in a mouse model of AS, our work does not rule out that the effect of LA on behavior could occur somewhere else beyond solely increasing PIEZO2 function in DRG neurons. Future experiments could be directed to test the effect of the LA diet on neuronal *Piezo2* conditional knockout mice. Nevertheless, our work provides proof of concept that manipulating fatty acid content in vivo can modulate the function of a mechanosensitive ion channel.

## Methods
### Ethics approval
**Animals.** Mice procedures described below were reviewed and approved by the University of Tennessee Health Science Center (UTHSC) Institutional Animal Care and Use Committee (UTHSC IACUC number: Protocol #19-0084). All methods were carried out in accordance with approved guidelines. Adult (2- to 4-month-old) mice were housed with a 12 h light/dark cycle at 21 °C with 40-60% humidity, and with food and water ad libitum. Male and female heterozygous *Ube3a*[m−/p+] maternal- and *Ube3a*[m+/p−] paternal-transmission knockout

mice were obtained from The Jackson Laboratory, strain C57BL/6 *Ube3a*[tm1Alb] (B6 AS; Stock No. 016590). C57BL/6 J mice (WT) were obtained from The Jackson Laboratory (Stock No. 000664). While the majority of mice used for these studies were purchased from the breeding colony maintained at JAX laboratories, we also bred mice from our in-house colony at UTHSC, and their genotype was confirmed using previously published protocols[5,96]. We used a two-step breeding scheme. Briefly, we first crossed female WT with male *Ube3a*[m+/p-] to generate heterozygous offspring, in which half of the progeny were *Ube3a*[m+/p-], with no AS phenotype. *Ube3a*[m+/p-] heterozygous females were crossed with WT males to generate *Ube3a*[m−/p+] mice, in which half of the progeny constitute AS mice displaying the corresponding phenotype. After confirming the genotype, the progenies were used for experiments. All experiments in this work were performed with male mice and their DRG neurons. The effect of the lack of *Ube3a* was also tested in DRG neurons from female mice (supplementary Fig. 1f-g).

**Control animals.** The *Ube3a* gene is imprinted in mammals, requiring separate crosses to generate *Ube3a*[m−/p+] vs. *Ube3a*[m+/p-] animals (as described above). We used both *Ube3a*[m−/p+] and *Ube3a*[m+/p-] animals on the same strain background generated at the same facility. Since we wanted to include the paternal inheritance of the *Ube3a*[-] allele (also as a control), it was not feasible to test animals from the same cross. Although many studies in the field use an F1 mixed BL6 control, we would argue that using animals in the same genetic background would be appropriate while only changing the inherited allele (maternal *vs.* paternal deficiency). The only disadvantage to our approach is that when comparing our data to laboratories that use a strict littermate approach, some behaviors may vary slightly. Nevertheless, we showed in our study that *Ube3a*[m−/p+] animals alone respond to a dietary intervention (same mice with the same genetic background). Figures in which experiments were performed with: *JAX laboratories mice*, Fig. 1a-e, Supplementary Fig. 1a-e, Fig. 7a-f, Supplementary Fig. 6b-d, Supplementary Fig. 7a. *In-house colony*, Supplementary Fig. 1f-k, Fig. 3e, Supplementary Fig. 5c-h, Supplementary Fig. 6e-i. *JAX laboratories and in-house*, Fig. 3a-d, Supplementary Fig. 3a-b, Fig. 6 a-e, Supplementary Fig. 5a-b, Supplementary Fig. 6a, Supplementary Fig. 7b-f, Fig. 8d-g, Supplementary Fig. 8a-e. *Littermates*, Supplementary Fig. 1f-g.

### Human cells
Neurotypical control teeth were obtained through the Department of Pediatric Dentistry at the UTHSC. Teeth from children with Angelman syndrome were collected remotely by the parents of these individuals under an approved Institutional Review Board (IRB) protocol. The UTHSC IRB approved this study, and informed consent was obtained for inclusion in the repository and experiments using de-identified DPSC-derived neurons. The differentiation of the de-identified DPSC lines into neurons for molecular studies is considered non-human subjects research under exemption #7 (II.111(a)(8)). The protected health information has been stripped from the biospecimens in the repository for secondary research studies, and each sample has been de-identified using a sample identifier. The human DPSCs used in this study are primary cell lines and, as a limited resource, are not available for wide distribution.

### Mouse DRG neurons
Primary cultures of mouse DRG neurons were obtained from 8–12-week-old male (unless otherwise noted) *Ube3a*[m−/p+] (maternal transmission) and *Ube3a*[m+/p−] (paternal transmission) mice. Mice were anesthetized with isoflurane and sacrificed by cervical dislocation. DRGs were dissected and kept on ice in 1X Hank's balanced salt solution (HBSS without $CaCl_2$ and $MgCl_2$). Next, DRGs were incubated with 1 mg/mL collagenase B (Sigma) in HBSS at 37 °C and 5% $CO_2$ and then, after 1 hour, were dissociated in a medium without serum. The cell suspension solution was centrifuged for 8 min at 62 $g$. The pellet was

resuspended in Dulbecco's modified Eagle medium (DMEM; Gibco) complete media containing 1% penicillin-streptomycin (Gibco), 1% MEM vitamin solution (Gibco), 1% L-glutamine (Gibco), and 10% horse serum (Gibco). Cells were plated on Poly-L-Lysine (Sigma-Aldrich)-treated glass coverslips in 24-well plates. All mouse DRG neurons were kept at 37 °C, with 95% relative humidity and 5% $CO_2$. Cells were used in experiments after 24-48 h.

## Cell culture

Mouse neuro-2a (N2A; catalog number CCL-131) cell line was obtained from the American Type Culture Collection (ATCC®). *PIEZO1* knock-out mouse N2A (*Piezo1*[−/−] N2A) cells were a gift from Dr. Gary R. Lewin. *Piezo1*[−/−] N2A cells were cultured in DMEM, 5% penicillin-streptomycin, and 10% fetal bovine serum (FBS). Human Merkel cell carcinoma cell line (MCC13; Cell Bank Australia reference number: CBA1338) was obtained from Sigma and cultured according to the manufacturer's protocol. MCC13 cells were cultured in RPMI1640 (with 2 mM L-glutamine and 25 mM HEPES; Sigma), 5% penicillin-streptomycin, and 10% FBS. All cultured cells were maintained at 37 °C, with 95% relative humidity and 5% $CO_2$. For actin experiments, cells were incubated in media supplemented with 1 μM latrunculin A or 0.5 μM jasplakinolide (Cayman Chemicals) for 18-24 h before recording or protein extraction.

## Cell transfections

*For electrophysiology*, we used Lipofectamine 2000 (ThermoFisher Scientific), according to the manufacturer's instructions. *N2A. Piezo1*[−/−] N2A cells were grown in 6-well plates to 80% confluency and co-transfected with 0.2 μg/ml of mouse *PIEZO2* variant 14-pcDNA3.1(+) and 50 ng/ml GFP-pMO (a pcDNA3.1-based vector with the 5′ and 3′ untranslated regions of the beta-globin gene) or transfected with 0.6 μg/ml MscL-pIRES2-EGFP for 48 h. The MscL construct was a gift from Dr. David E. Clapham. *MCC13.* For electrophysiological recordings, MCC13 cells were grown in 6-well plates to a 90% confluency and co-transfected with 1.1 μg/ml of human *UBE3A* (pLVX-HA-UBE3A-IRES-Puro; a gift from Robert Sobol; Addgene plasmid # 107349)[97] or 0.45 μg/ml of human cofilin (pEGFP-N1 human *Cofilin* WT; a gift from James Bamburg; Addgene plasmid # 50859)[98] and 150 ng/ml GFP-pMO for 24 h. For protein expression assays, MCC13 cells were grown in 10 cm Petri dishes to a confluency of ~100% and transfected with 1.1 μg/ml of human *UBE3A* or 1 μg/ml of the empty vector pMO for 24 h.

*For pull-down assays*, HEK293T cells were grown at 37 °C, with 95% relative humidity and 5% $CO_2$, in DMEM/F-12 with GlutaMAX (Thermo Scientific) supplemented with 10 % FBS (Thermo Scientific), 100 U/ml of penicillin (Invitrogen), and 100 μg/ml of streptomycin (Invitrogen). A total of 1 ×10^6 cells were seeded in six-well plates for GFP pulldown experiments. On the next day, cells were co-transfected with 0.75 μg of *Cofilin-GFP* plasmid, 0.75 μg of *FLAG-Ubiquitin* plasmid[99], and 0.5 μg of human wild-type *UBE3A* isoform 2, catalytically inactive *UBE3A* isoform 2, or *pmCherry-N1* plasmid (Clontech) as a control. 48 h later cells were harvested and subjected to a GFP pull-down assay.

## siRNA-mediated knockdown and real-time quantitative PCR

MCC13 cells (grown to ~100% confluency), primary cultures of mouse DRG neurons, and human iPSC-derived neurons were transfected with Lipofectamine® RNAiMAX Transfection Reagent (ThermoFisher Scientific), according to the manufacturer's protocols. For electrophysiology experiments, the siRNA concentration used for MCC13 cells and iPSC-derived neurons was 20 nM for *UBE3A*, *PIEZO2*, or the silencer negative control (scrambled siRNA; Ambion, Inc) and 60 nM for protein expression assays. For mouse DRG neurons, the siRNA concentration was 60 nM for *Cofilin* (AA959946, Horizon Discovery Ltd.) or the silencer negative control. The transfections were done with antibiotic-free media. After 6 h of transfection, the medium was

replaced with fresh media containing antibiotics. All cells were used 48 h after transfection. For electrophysiology experiments, cells were also co-transfected with siGLO Green Transfection Indicator (Dharmacon). Human *UBE3A* was silenced with a validated siRNA (Thermo-Fisher Scientific #4390825). The targeted sequences of siRNAs directed against human *PIEZO2* RNA (Dharmacon) were:

5′-UCGAAAGAAUAUCGCUAAA-3′;         5′-UCGGAAAUAGCAACA-GAUA-3′;

5′-GGAACUAAUUGCCCGUGAA-3′; and 5′-CUAUGGUAUUAUGG-GAUUA-3′.

RNAs were isolated from MCC13 cells using RNeasy® Mini Kit (QIAGEN). cDNA was generated with iScript™ Reverse Transcription Supermix for RT-qPCR (Bio-Rad). qPCR triplicate reactions were run in a CFX Connect™ Real-Time PCR detection system with SsoAdvanced™ Universal SYBR® Green Supermix (Bio-Rad), according to the manufacturer's instructions. Data acquisition and analyses were made using Bio-Rad CFX Maestro 1.0 (Bio-Rad Laboratories). Normalization was done using ΔΔCq, with GAPDH as the reference gene. Standard curves were generated from specific templates for each PrimePCR™ assay listed below. Primers were purchased from Bio-Rad for Human *GAPDH* (Unique Assay ID: qHsaCED0038674), *UBE3A* (Unique Assay ID: qHsa-CID001 6100), and *PIEZO2* (Unique Assay ID: qHsaCID00 06733).

## Lipid supplementation

Pure (>99%) linoleic acid (LA), alpha-linolenic acid (αLA), gamma-linolenic acid (γLA), dihomo gamma-linolenic acid (DγLA), arachidonic acid (AA), docosahexaenoic acid (DHA), and docosatetraenoic acid (DTA) were obtained from Nu-Chek Prep, Inc. Ampoules were opened and their content diluted in DMSO (Sigma-Aldrich) ≤ 20 min before the addition of the PUFAs to the corresponding culture media (media was warmed in advance to 37 °C). Control conditions include only DMSO in the cell culture media. PUFA-enriched media was placed in a water bath (37 °C) for 5-10 min and vortexed three times until no visible lipid aggregates were observed. The supplementation protocols described below were optimized for cell survival and quality. We observed some cell death (10-30%) when LA supplementation was ≥ 200 μM.

*Piezo1*[−/−] *N2A cells* grown in 6-well plates were supplemented with PUFA-enriched media 18-24 h after transfection. 18 h later (i.e., 48 h after transfection), the cells were trypsinized and plated on Poly-L-Lysine (Sigma-Aldrich)-treated glass coverslips in the corresponding media, without supplemented fatty acids. After plating, the cells were given 3 h to attach before electrophysiological recording (cells plated for more than 12 h were not used). *MCC13 cells* grown to 80-90% confluency in 6-well plates (no floating cells visible) were supplemented with LA-enriched media for 18-24 h. Next, cells were trypsinized and plated on Poly-L-Lysine-treated glass coverslips in the corresponding media, without LA supplementation. After plating, the cells were given 3 h to attach before electrophysiological recordings. *Primary mouse DRG neurons* were supplemented with LA using a two-day protocol, which improved neuronal viability and the ability to perform patch-clamp recordings. DRGs were plated on Poly-L-Lysine-treated glass coverslips in a 24-well plate. 4-6 h later, the culture was supplemented with media containing 50 μM LA for 24 h and then exchanged for fresh media with 100 μM LA for another 24 h (48 h total) before electrophysiological experiments.

*Human iPSC-derived neurons* were plated on glass coverslips pretreated with Poly-L-Lysine and Matrix 3, according to the manufacturer's instructions (Anatomic Incorporated, #M8003). Thirteen days later, iPSC-derived neurons were supplemented with LA using the two-day protocol.

*Human DPSC-derived neurons* were plated on glass coverslips pretreated with Poly-D-lysine and Matrigel (Corning). Thirty days later, DPSC-derived neurons were supplemented with LA using the two-day protocol.

## Electrophysiology

Patch-clamp recordings were performed on cells plated on glass coverslips. For whole-cell recordings of mechano-activated currents in the voltage-clamp mode, the bath solution contained 140 mM NaCl, 6 mM KCl, 2 mM CaCl$_2$, 1 mM MgCl$_2$, 10 mM glucose, and 10 mM HEPES (pH 7.4), while the pipette solution contained 140 mM CsCl, 5 mM EGTA, 1 mM CaCl$_2$, 1 mM MgCl$_2$, and 10 mM HEPES (pH 7.2). For whole-cell recordings of TRP channels, the bath solution contained 140 mM NaCl, 6 mM KCl, 2 mM MgCl$_2$, 10 mM glucose, and 10 mM HEPES (pH 7.4), while the pipette solution contained 140 mM CsCl, 5 mM EGTA and 10 mM HEPES (pH 7.2). For inside-out recordings, symmetrical conditions of 140 mM KCl, 6 mM NaCl, 2 mM CaCl$_2$, 1 mM MgCl$_2$, 10 mM glucose, and 10 mM HEPES (pH 7.4) were used. For current-clamp recordings of action potentials elicited by mechanical stimulation, the bath solution contained 140 mM NaCl, 6 mM KCl, 2 mM CaCl$_2$, 1 mM MgCl$_2$, 10 mM glucose, and 10 mM HEPES (pH 7.4), while the pipette solution contained 140 mM KCl, 6 mM NaCl, 2 mM CaCl$_2$, 1 mM MgCl$_2$, 10 mM glucose, and 10 mM HEPES (pH 7.4). For current-clamp recordings of action potentials elicited by current injection, the bath solution contained 125 mM NaCl, 3 mM KCl, 1 mM MgCl$_2$, 2.5 mM CaCl$_2$, 10 mM HEPES, 10 mM glucose (pH 7.4), while the pipette solution contained 130 mM KCl, 4 mM Na$_2$ATP, 3 mM MgCl$_2$ and 10 mM HEPES (pH 7.3). Pipettes were made of borosilicate glass (Sutter Instruments) and fire-polished to a resistance between 3 and 5 M$\Omega$ before use. Mechanical stimulation was performed using the voltage-clamp (constant −60 mV) or current-clamp configuration. Recordings were sampled at 100 kHz and low pass filtered at 10 kHz using a MultiClamp 700 B amplifier and Clampex (v10.4.2.0; Molecular Devices, LLC). Leak currents before mechanical stimulation were subtracted offline from the current traces and data were digitally filtered at 2 kHz with ClampFit (v10.4.2.0; Molecular Devices, LLC). Recordings with leak currents >200 pA, with access resistance >10 M$\Omega$, and cells with giga-seals that did not withstand at least five consecutive steps of mechanical stimulation were excluded from analyses. For human DPSC-derived neurons, cells with giga-seals, notwithstanding at least four consecutive steps of mechanical stimulation, were excluded.

## Mechanical stimulation

For indentation assays, DRG neurons, as well as MCC13, N2A, *Piezo1*$^{-/-}$ N2A, iPSC- and DPSC-derived cells, were mechanically stimulated with a heat-polished blunt glass pipette (3–4 µm) driven by a piezo servo controller (E625, Physik Instrumente). The blunt pipette was mounted on a micromanipulator at an ~45° angle and positioned 3–4 µm above the cells. Displacement measurements were obtained with a square-pulse protocol consisting of 1 µm incremental indentation steps, each lasting 200 ms with a 2-ms ramp in 10-s intervals. The threshold of mechano-activated currents for each experiment was defined as the indentation step that evoked the first current deflection from the baseline. For current-clamp experiments, the mechanical threshold was defined as the indentation step that produced the first action potential. Only cells that did not detach throughout stimulation protocols were included in the analyses. The piezo servo controller was automated, using a MultiClamp 700B amplifier, through Clampex (Molecular Devices, LLC). For pressure-clamp assays, excised membrane patches were mechanically stimulated with negative pressure applied through the patch pipette using a High-Speed Pressure Clamp (ALA Scientific), which was automated, using a MultiClamp 700B amplifier, through Clampex. Inside-out patches were probed using a square-pulse protocol consisting of a 50 ms 10 mmHg pre-pulse, immediately followed by incremental (−10 mmHg) pressure steps, each lasting 200 ms in 1-s intervals.

## Protein expression determination

Transfected, Latrunculin A (1 µM; 20–24 h)-treated, or LA-supplemented MCC13 cells were treated with ProteoExtract®

Subcellular Proteome Extraction Kit (Millipore Sigma) to obtain cytosolic, membrane, and cytoskeletal fractions, according to the manufacturer's instructions and elsewhere[100]. DRGs from WT and *Ube3a*$^{m-/p+}$ mice were treated with the G-actin / F-actin In Vivo Assay Kit (Cytoskeleton, Inc.) to obtain globular (G) and filamentous (F) actin forms, according to the manufacturer's instructions. Protein concentrations were measured with the Bio-Rad protein assay or the Pierce™ BCA Protein Assay Kit (ThermoFisher Scientific), and equivalent protein amounts were loaded in Mini-PROTEAN TGX Stain-Free Precast Gels (Bio-Rad). Mouse monoclonal anti-UBE3A (1:1,000; Sigma-Aldrich Cat# SAB1404508, RRID:AB_10740376), mouse monoclonal anti-actin for MCC13 cells (1:5,000; Bio-Rad Cat# MCA358GT, RRID:AB_323521), mouse monoclonal anti-actin for DRGs (1:1,000; Cytoskeleton Cat# AAN02, RRID:AB_2884962), goat polyclonal anti-mouse IgG H&L (HRP) (1:20,000; Abcam Cat# ab205719, RRID:AB_2755049), rabbit polyclonal anti-human PIEZO2 (1:1,000; abcepta Cat# AP16313b, RRID:AB_11136435), rabbit polyclonal anti-cofilin (1:1,000; abcepta Cat# AP53892, RRID:AB_2923306), rabbit monoclonal anti-cofilin for DRGs (1:1,000; Cell Signaling Technology Cat# 5175, RRID:AB_10622000), and goat anti-rabbit IgG HL-HRP conjugated (1:10,000; Bio-Rad Cat# 1706515, RRID:AB_2617112) antibodies were used for western blots. Membranes were developed with SuperSignal™ West Femto Maximum Sensitivity Substrate and imaged in a ChemiDoc Touch Imaging System (Bio-Rad) for chemiluminescence. Western blots were analyzed, using Image Lab Software (v6.1.0; Bio-Rad), by measuring each chemiluminescent signal *vs.* the total protein loaded (stain-free signal) in the corresponding lane[101,102]. We reported band relative intensities, and for experiments utilizing MCC13 cells, we normalized against the control treatment.

## Immunostaining

Cultured DRG neurons were fixed with 3% paraformaldehyde and 0.1% glutaraldehyde in PBS for 20 min and permeabilized with 0.1% Triton X-100 in PBS for 10 min. DRG neurons were then incubated with a high-affinity F-actin probe, Alexa Fluor 488™ phalloidin (1:40, Thermo-Fisher Scientific # A12379), in PBS for 20 min at RT, followed by a 5 min incubation with the nuclear counterstain DAPI (ThermoFisher Scientific # 62248). Samples were preserved using ProLong Antifade Mountant (ThermoFisher Scientific # P36961). Images were acquired on a Zeiss LSM710 confocal microscope with a 63x/1.4 Plan-Apochromat oil immersion objective. Images were processed using ZEN2010 software (Zeiss) and analyzed using Fiji ImageJ[103] (v.2.3.0/1.53q) to enhance contrast and convert to an appropriate format.

## Cofilin ubiquitination assay

**GFP pulldowns.** GFP pulldowns were performed as previously described[46,47,99,104]. Briefly, cells were lysed with 300 µl of lysis buffer and centrifuged at 14,000 g at 4 °C for 10 min. Supernatants were then diluted with 450 µl of dilution buffer, except 30 µl that were kept as input fractions. Afterward, diluted supernatants were incubated with 20 µl of GFP-Trap-A agarose bead suspension (Chromotek GmbH) for 2 h and 30 min at room temperature, with gentle rolling. GFP beads were subsequently washed once with 1 ml of dilution buffer, three times with 1 ml of washing buffer, and once with SDS buffer. Bound GFP-tagged Cofilin was eluted by boiling the GFP beads at 95 °C for 10 min in 20 µl of elution buffer. Finally, samples were centrifuged at 16,000 g for 2 min to get rid of the GFP-Trap-A agarose beads. Buffer compositions were as follows: lysis buffer − 50 mM Tris-HCl, pH 7.5, 150 mM NaCl, 1 mM EDTA, 0.5 % Triton, 50 mM N-ethylmaleimide (Sigma), and 1X protease inhibitor cocktail (Roche); dilution buffer − 10 mM Tris-HCl, pH 7.5, 150 mM NaCl, 1 mM EDTA, 50 mM N-ethyl-maleimide, and 1X protease inhibitor cocktail; washing buffer − 8 M urea, 1 % SDS, 1X PBS; SDS buffer, 1 % SDS, 1X PBS; elution buffer − 250 mM Tris- HCl, pH 7.5, 40 % glycerol, 4 % SDS, 0.2 % bromophenol blue, and 100 mM DTT. *Western blots.* Input and elution fractions from

GFP pulldowns were resolved by SDS PAGE, using 4-12 % Bolt Bis-Tris Plus pre-cast gels (Invitrogen). Proteins were then transferred to PVDF membranes using the iBlot system (Invitrogen). Following primary and secondary antibody incubation, membranes were developed by either chemiluminescence, using the Clarity Western ECL Substrate kit (Bio-Rad), or near-infrared fluorescence. The following primary antibodies were used: mouse monoclonal anti-GFP antibody (1:1,000, Roche Cat# 11814460001, RRID:AB_390913), mouse monoclonal anti-FLAG M2-HRP conjugated antibody (1:1,000, Sigma-Aldrich Cat# A8592, RRID:AB_439702), and mouse monoclonal anti-UBE3A (1:1,000, clone E6AP-300) antibody (1:1,000, Sigma-Aldrich Cat# E8655, RRID:AB_261956). The following secondary antibodies were used: goat anti-mouse-HRP-labeled antibody (1:4,000; ThermoFisher Scientific Cat# 62-6520, RRID:AB_2533947) and goat anti-mouse IRDye-800CW (1:8,000, LI-COR Biosciences Cat# 926-32210, RRID:AB_621842).

### Liquid chromatography-mass spectrometry
Control and PUFA-treated *Piezo1*$^{-/-}$ N2A cells (2−4 ×10$^6$ cells), as well as mouse DRGs dissected from mice fed with a standard or LA-enriched diet, were rinsed with PBS three times and flash-frozen in liquid N$_2$. Total and free fatty acids were extracted and quantified at the Lipidomics Core Facility at Wayne State University. Membrane (i.e., esterified) fatty acids were determined by subtracting free fatty acids from total fatty acids and normalized by cell density or total protein.

### Differential scanning calorimetry and data analysis
A stock solution of 1,2-dipalmitoyl-sn-glycero- 3-phosphocholine (DPPC, Avanti Polar Lipids) was prepared in ethanol (4 mM). DPPC solutions were aliquoted, distributed into glass tubes, and mixed with the respective fatty acid (FA) at 5 mol%. The solvent was evaporated from lipid mixtures under a N$_2$ stream while stirring at 55 °C for 40 min. DPPC/FA films were hydrated with 10 mM HEPES (pH 7.4) at ~55 °C, followed by vigorous vortex mixing. To yield multilamellar vesicles, mixtures were softly stirred at 600 rpm for 40 min at 55 °C using a Degassing Station (TA Instruments). Samples were degassed at low pressure (635 mmHg) for 10 min at 25 °C before loading into the DSC capillaries. Heat capacity profiles were recorded on a NanoDSC microcalorimeter (v4.8.2; TA Instruments) at a constant scan rate of 1 °C/min (between 25 and 50 °C) and a constant pressure of 3 atm. Samples were equilibrated for 5 min at 25 °C before each experiment.

Single thermograms were analyzed with the NanoAnalyze software (v3.12.0; TA instruments). To evaluate the extent of variation of the lipid organization, we determined the cooperative unit 'κ' of the main-phase transition of lipid systems using the following equation[105]:

$$\kappa = \Delta H_{vH}/\Delta H_{cal} \qquad (1)$$

where $\Delta H_{cal}$ corresponds to the experimental enthalpy variation provided by the area under the curve of the main transition peak. $\Delta H_{vH}$ indicates the van't Hoff enthalpy change, an estimate of the enthalpy associated with a simple two-state first-order transition model, which is given by:

$$\Delta H_{vH} = 6.9 \frac{J}{K\,mol} \cdot \frac{T_m^{\;2}}{\Delta T_{1/2}} \qquad (2)$$

where $T_m$ is the melting transition temperature, and $\Delta T_{1/2}$ corresponds to the width at half-height of the transition peak. Calculations were performed through an optimized code on MATLAB (signal processing toolbox).

### Diet supplementation
WT, *Ube3a*$^{m-/p+}$ (maternal transmission), and *Ube3a*$^{m+/p-}$ (paternal transmission) mice (6-8 weeks old) were pair-fed for 8-12 weeks (before DRG dissection or behavioral experiments) with a: **A)** standard diet (sd;

#7012 ENVIGO; UTHSC animal facility diet; (https://insights.envigo.com/hubfs/resources/data-sheets/7012-datasheet-0915.pdf); **B)** Safflower oil-supplemented diet (enriched in LA, Dyets # 112245). Modified AIN-93G purified rodent diet with 59% fat derived calories from safflower oil (kcal/kg): Casein (716), L-Cystine (12), Maltose Dextrin (502), Cornstarch (818.76), Safflower Oil (2430), Soybean Oil (630), Mineral Mix (# 210025; 30.8), and Vitamin Mix (#310025; 38.7); **C)** High-fat diet (anhydrous milk fat supplemented, Dyets # 105012). Modified AIN-93G Purified Rodent Diet with 59% Fat Derived Calories from Anhydrous Milk Fat (kcal/kg): Casein (716), L-Cystine (12), Maltose Dextrin (502), Cornstarch (818.76), Anhydrous Milk Fat (2430), Soybean Oil (630), Mineral Mix (# 210025; 30.8), and Vitamin Mix (#310025; 38.7).

### Behavioral analysis
Behavioral analyses were evaluated in the C57BL/6 J, *Ube3a*$^{m-/p+}$ maternal-, and *Ube3a*$^{m+/p-}$ paternal-transmission knockout adult mice fed with a standard, LA-enriched, or high-fat diet (anhydrous milk fat supplemented). *Catwalk*. Gait dynamics were quantified using the CatWalkXT (v10.0.408; Noldus, USA). Mice were first acclimated to a dark room and the CatWalk instrument over five days for 1 h with the illuminated surface turned on. Each mouse could walk freely in both directions with a minimum level of disturbing external elements. The CatWalk system automatically recorded video of the mice walking the entire length of the walkway. Experimental sessions typically lasted 10 min, and three compliant runs were recorded per mouse. The CatWalk system determined the compliance (60%) according to the run's duration and speed variation. Paw positions were verified manually after acquisition.

**Tail-clip.** Response to noxious pressure application was determined with the tail-clip assay. Mice were acclimated for 5 min in von Frey chambers. A blunt alligator clip was applied to the middle of the tail and mice were immediately placed back into the von Frey chamber. The latency to lick, bite or grab was scored with a stopwatch. A maximum cut-off latency of 60 s was imposed to prevent tissue damage. Time to respond was reported as the latency to react for each mouse. Data was analyzed in Excel (v:2211; Microsoft Corp.) and plotted using OriginPro 2018 (v:b9.5.1.195; OriginLab Corp.).

**Hot plate.** Mice were placed on a heated metal surface (Ugo Basile) maintained at 52 ± 0.5 °C by a feedback-controlled Peltier device (Harvard Apparatus). The latency time between placement and the nociceptive behavior (licking, biting, lifting, guarding, shaking the hind paws, or jumping) was scored. The animal was immediately removed from the apparatus after showing a response. For acclimation, each animal was placed in the inactive chamber for 45 min, one day before the test. A cut-off time of 30 s was established to avoid tissue injury. Data was analyzed in Excel (v:2211; Microsoft Corp.) and plotted using OriginPro 2018 (v:b9.5.1.195; OriginLab Corp.).

**Hargreaves test.** Paw withdrawal latencies were determined by testing left and right hind paw responses to radiant heat. Latency time to the onset of one of the following nocifensive behaviors was measured: licking, biting, lifting, guarding, or shaking the hind paw. Once a score was determined, the heating was automatically switched off. For acclimatization, mice were placed into the chamber for 1 hour each of the 5 days preceding the test. A cut-off time of 20 s was established to avoid tissue injury. Four measurements were taken per mouse, and the average for each mouse was reported. Data was analyzed in Excel (v:2211; Microsoft Corp.) and plotted using OriginPro 2018 (v:b9.5.1.195; OriginLab Corp.).

### Mouse cytokine profile assay
Mouse blood was obtained by cardiac puncture under general anesthesia, and immediately mixed with a 10% vol. of heparin (1000 units/

ml; Sagent Pharmaceuticals), and spun down at 2,000 g for 15 min[106]. Then, the plasma was separated and used for the cytokine profile assay. The cytokine assay was performed with the Mouse Cytokine Array Panel kit (R&D Systems, Inc., #ARY006). Briefly, the plasma sample was diluted and mixed with a detection antibody cocktail. The mixture was incubated overnight with the mouse cytokine assay membrane at 4 °C. Streptavidin-HRP was added after washes to remove unbound materials, and a chemiluminescent reagents mix was used for detecting cytokine binding. Membranes were imaged in a ChemiDoc Touch Imaging System (Bio-Rad) for chemiluminescence.

### Human iPSC-derived sensory neurons
The human iPSC line was manufactured and characterized at Anatomic Incorporated facilities, with informed consent, using proprietary technologies. hiPSCs were differentiated into sensory neurons using a commercially available kit, Senso-DM, according to the manufacturer's instructions (Anatomic Incorporated, #7007). Briefly, undifferentiated human induced pluripotent stem cell line #1 (hiPSC; from female donor's umbilical cord blood) was maintained with TeSR™-E8™ (STEMCELL Technologies Inc., 05990) on tissue culture-treated plastic that was coated with truncated recombinant human vitronectin (ThermoFisher Scientific # A14700). hiPSCs were dissociated using EDTA (0.55 mM) and single cell-seeded into TeSR-E8 supplemented with 10 μM Y-27632 (Selleckchem S1049) in flasks pre-coated with diluted 1:100 Matrix 1 in dPBS with Ca²⁺ and Mg²⁺ (ThermoFisher Scientific # 14040-133) to induce development into primal ectoderm[68]. Subsequent daily media changes of Senso-DM 1-7 generated a population of immature sensory neurons, which were cryopreserved and later thawed for studies. Sensory neurons were maintained in culture at 37 °C with 5% CO₂ on glass coverslips coated with Poly-L-Lysine and Matrix 3 (Anatomic Incorporated, #M8003) in Senso-MM media (Anatomic Incorporated, #7008). Human iPSC-derived sensory neurons were incubated in culture for 14 days before recording. For siRNA-mediated knockdown assays, sensory neurons were co-transfected with siGLO Green Transfection Indicator (Dharmacon) and siRNA for *UBE3A*, *PIEZO2*, or the silencer negative control (scrambled siRNA; described above), using the Lipofectamine® RNAiMAX Transfection Reagent (ThermoFisher Scientific), according to the manufacturer's protocols, 24 h before electrophysiological recording. The transfections were performed using antibiotic-free media.

### Generation of dental pulp stem cell (DPSC) cultures and neuronal differentiation
Deciduous teeth were either collected locally, through the Department of Pediatric Dentistry (neurotypical controls; n = 2), or remotely, for children with AS (n = 6). Immediately following the loss of the tooth, it was placed in transportation media (DMEM/F12 50/50 mix with HEPES, 100 U/mL penicillin, and 100 μg/mL streptomycin). DPSCs were isolated and cultured according to our previously described protocol[72]. Briefly, after mincing the dental pulp from inside the tooth cavity, 3 mg/mL dispase II and 4 mg/mL collagenase I were added to digest the tissue. Cells were then seeded on Poly-D-Lysine-coated 12-well plates with media containing DMEM/F12 1:1, 20% FBS, newborn calf serum (NCS), and 100 U/mL penicillin and 100 μg/mL streptomycin (Pen/Strep). Confluent cultures (80%) were passaged with TrypLE Express (Gibco), and neuronal differentiation was performed only on early passage cells (less than four).

DPSC lines were seeded at 20,000 cells/cm² on Poly-D-Lysine and Matrigel-coated coverslips with DMEM/F12 1:1, 10% FBS, 10% NCS, and Pen/Strep. At 80% confluence, the neuronal differentiation protocol was followed as previously published[71]. Briefly, epigenetic reprogramming was performed by exposing the DPSC to 10 μM 5-azacytidine (Acros Scientific) in DMEM/F12 containing 2.5% FBS and 10 ng/mL bFGF (Fisher Scientific) for 48 h. Neural differentiation was induced by exposing the cells to 250 μM IBMX, 50 μM forskolin,

200 nM TPA, 1 mM db-cAMP (Sigma Aldrich), 10 ng/mL bFGF, 10 ng/mL NGF (Invitrogen), 30 ng/mL NT-3 (Peprotech), and 1% insulin-transferrin-sodium selenite premix (ITS) (Fisher Scientific) in DMEM/F12 for 3 days. At the end of neural induction, the cells were washed with 1X PBS. Neuronal maturation was performed by maintaining the cells in Neurobasal A media (Gibco) with 1 mM db-cAMP, 2% B27, 1% N2 supplement (Gibco), 30 ng/mL NT-3, and 1X Glutamax (Gibco) for 30 days.

### Data analysis
Data were plotted using OriginPro (2018 v:b9.51.195; OriginLab Corp.) and Estimation Stats[107]. The time constant of inactivation τ was obtained by fitting a single exponential function *(3)* between the peak value of the current at the end of the stimulus:

$$f_{(t)} = \sum_{i=1}^{n} A_i * e^{-t/\tau_i} + C \qquad (3)$$

where A = amplitude, τ = time constant, and the constant y-offset *C* for each component *i*.

Sigmoidal fitting was done using OriginPro with the following Boltzmann function:

$$f_{(x)} = A_2 + \frac{A_1 - A_2}{1 + e^{((X - X_o)/dX)}} \qquad (4)$$

where $A_2$ = final value, $A_1$ = initial value; $X_o$ = center, and $dX$ = time constant.

All boxplots show mean (square), median (bisecting line), bounds of box (75th to 25th percentiles), and outlier range with 1.5 coefficient (whiskers). Statistical analyses were performed using GraphPad InStat software (version 3.10; GraphPad Software Inc.) and Estimation Stats[107]. We used the Kolmogorov and Smirnov method to determine data distribution, as well as the Bartlett's test to determine differences between standard deviations. Individual tests are described in each of the Fig. legends.

### Statistics & Reproducibility
No statistical method was used to predetermine the sample size. No data were excluded from the analyses. The experiments were not randomized. For electrophysiology, the investigator was blind to genotype and treatment. For behavioral assays, the investigators were blind when possible. All attempts at replication were successful. Experiments were performed at least three times on different days with different/independent preparations.

### Cartoon images
Figure 7b (mouse), Fig. 8a (mouse), Fig. 8b (catwalk setup), Fig. 10 (model), supplementary Fig. 7b-e (behavioral test setups), and supplementary Fig. 8e (rotarod) were created with bioRENDER.com.

### Reporting summary
Further information on research design is available in the Nature Portfolio Reporting Summary linked to this article.

## Data availability
The source data underlying Figures and Supplementary Figures are provided as a Source Data file available at Figshare. https://doi.org/10.6084/m9.figshare.20038184 Source data are provided with this paper.

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

## Acknowledgements

We thank Dr. A.T. Chesler and Dr. M.A. Puchowicz for their experimental advice. We thank Dr. T. Ishrat and Dr. S. Ismael for access to and technical support for the CatWalk instrument, Dr. M. Ennis for access to the Hargreaves test apparatus, and Dr. L.M. Pfeffer for access to the Bio-Rad CFX qPCR. We thank Dr. G.R. Lewin for providing *Piezo1−/−* N2A cells and Dr. D.E. Clapham for MscL-pIRES2-EGFP. We thank M.Sc. B. Bell, Dr. A.T. Chesler, Dr. R.C. Foehring, Dr. M.A. Puchowicz, and Dr. G. Tigyi for critically reading the manuscript, and Graduate Student M. Bade and Dr. R. Kumar for technical assistance. We utilized the lipidomics Core Facility at Wayne State University (NIH S10RR027926). This work was supported by the Neuroscience Institute at UTHSC (Research Associate Matching Salary Support to J.L.), the Federico Baur endowed chair in Nanotechnology (to F.J.S.-V., 0020206BA1), a pilot research award from the Foundation for Prader-Willi Research (to L.T.R.), the Neuroscience Institute Research Supports Grant 2020 program (to V.V., and J.F.C.-M.), and the National Institutes of Health (R01GM133845 to V.V. and R01GM125629 to J.F.C.-M.).

## Author contributions

Lead authors, J.F.C.-M and V.V.; Conceptualization: V.V. and J.F.C.-M. Methodology: L.O.R., R.C., U.M., L.T.R., V.V., and J.F.C.-M. Formal Analysis: L.O.R, R.C., V.V., and J.F.C.-M. Investigation: L.O.R, R.C., A.K.V., J.R., P.W., V.T. F.J.S.-V., J.L., and V.V. Writing-original draft: L.T.R., V.V. and J.F.C.-M; Review and editing: L.O.R, L.T.R., V.V. and J.F.C.-M. Supervision: L.T.R., V.V. and J.F.C.-M.

## Competing interests

P.W. and V.T. are employees and shareholders of Anatomic Incorporated. There are no other competing interests.
