## [Peer Review File · Nature Communications]

Linoleic Acid Improves PIEZO2 Dysfunction in a mouse model of Angelman SyndromeReviewer #1 (Remarks to the Author):

Romero et al studies the role of Piezo2 channel in the development of ataxia in Angelman syndrome. Angelman syndrome is caused by loss of function of maternally derived UBE3A, a ubiquitin ligase. Angelman syndrome causes delayed development, intellectual disability as well as ataxia. The ataxic phenotype is attributed to cerebellar disorder, but there are data indicating that cerebellar dysfunction is only partially responsible for ataxic gait. Piezo2 is a mechanically activated ion channel, and the major symptom of its loss of function mutations in humans is ataxia.

The manuscript presents three major findings. 1. The authors show that in a mouse model of Angelman syndrome (Ube3a maternal KO), as well as in stem cell-derived neurons from Angelman syndrome patients, mechanically activated currents in DRG neurons are decreased. Ube3a knockdown in a Merkel cell carcinoma line also decreased Piezo2 mediated currents. 2. Linoleic supplementation increased Piezo2 mediated currents in Ube3A deficient (and wild type) cells, and alleviated the gait problems (increased stride length) in a mouse model of Angelman syndrome. 3. The authors propose a mechanism in which lack of Ube3a causes reduced ubiquitination of cofilin, leading to decreased actin levels which results in reduced Piezo2 currents.

I think the manuscript is important, as it addresses the mechanism of some of the symptoms of a devastating disease. The authors propose a feasible pathomechanism, as well as a potential therapy that is potentially very simple to implement. The data are solid, rigorous and convincing, and I strongly support publication after addressing the comments below.

1. Neither the effect of Ube3a reduction nor the effects linoleic acid are specific to Piezo2 or to rapidly adapting mechanically activated currents in DRG neurons. The logical link between Piezo2 and the ataxic phenotype of Angelman syndrome is provided by the knowledge that Piezo2 loss of function mutations in humans cause ataxia. Do children with Angelman syndrome have reduced light touch ? If so, please discuss. Reduced light touch is also a symptom of Piezo2 loss of function mutations in humans. Also, do Ube3a knockout mice have other signs of Piezo2 deficiency at the behavioral level, such as reduced sensitivity in the von Frey assay ?

2. Figure 2 siRNA experiments. The effect of siRNA on the Western is quite small, the effect on currents is a lot clearer. Was there a fluorescent marker for transfection used ? If so, please state, as it can explain the difference between the effects size on the currents and the Western blot, assuming that only fluorescent cells were patched, and transfection efficiency was not 100%. The gel pictures shows suggest a more than 50% reduction in intensity, but the summary figure shows ~50 % as the largest reduction. I would double check the numbers.

Also, the significance between control and siRNA treatment is not noted. I think that is a more relevant comparison than the lack of significant difference between Piezo2 siRNA and Ube3a siRNA. The same applies for panels f,g,h, where fold changes are shown, significance should be show, one sample t-test, difference from 1, or paired t-test between individual values would suffice, provided that the data are normally distributed.

Minor comments:

3. I would show exact p values including with those labeled ns. In most cases the effects are very clear, but in some panels such as Figure 4 b,c there are some visible trends in some of the "ns" panel, and the actual effect is only one asterisk, so exact p values would be informative.

4. Some of the statistical comparisons were tested with parametric test. i.e. t-test and ANOVA, some with non-parametric. Please explain if this was based of testing normality of data distribution.

5. Title: I think Angelman syndrome denotes the human condition. I would change the title to "...mouse model of Angelman Syndrome", or similar. Also the title describe a subset of findings, the authors may consider a title that is more inclusive of the rest of the findings.

6. Abstract, I would clarify that Piezo2 is a mechanically activated ion channel

Reviewer #2 (Remarks to the Author):

Angelman syndrome (AS) is a neurogenetic disorder characterized by intellectual disability and movement deficits that arises as a consequence of loss of expression of UBE3 E3 ligase from the maternal allele. Given that both individuals with AS and those with loss of function of PIEZO2 display impaired balance, coordination and gait, the authors here set out to determine whether sensory neuron PIEZO2 is attenuated in AS and conditions in which UBE3 expression is suppressed. They found that PIEZO2 currents are significantly reduced in sensory neurons from a mouse model of AS, in cells in which UBE3 is knocked down, as well as human stem-cell derived neurons from AS patients. They link the decrease in PIEZO2 to a decrease in F-actin that is downstream of enhanced expression of cofilin, an actin binding protein that promotes actin disassembly. They further found that the deficits in PIEZO2 in neurons from AS mice or derived from human stem cells can be reversed by incubation with linoleic acid (LA) which acts by modifying the membrane properties in a manner that is sensed by PIEZO2 but not other membrane channels. Finally, they showed that AS mice fed with LA-enriched diet displayed normalized PIEZO2 currents and improved gait.

Overall, this study is impressive in its scope and translational potential. The experiments are technically well done and much of the interpretation is supported by the data. There are some issues that need to be addressed to strengthen some of the conclusions.

1. One area in which the manuscript is unclear is in the mechanism of action of linoleic acid in selectively enhancing PIEZO2 currents. The authors suggest that this is due to membrane remodeling, which is a rather non-specific effect that might be expected to alter the properties of multiple membrane ion channels which is not observed. Do none of the other PUFAs tested in Fig. 4 (which are not effective in increasing PIEZO2 currents) change membrane properties? Unfortunately, the DSC experiments conducted for LA (Supplemental Figs. 4i and 4j) were not done for these other PUFAs to address this question.

2. The study utilizes a variety of cell types – DRG neurons, MCC cells, N2A cells, stem-cell-derived neurons. On the one hand, it is impressive and reinforcing that the correlation between UBE3 expression and PIEZO2 current amplitude persists across all these different cell types. On the other hand, I don't think the use of all of them is sufficiently justified in the manuscript and, in my view, they detract from the overall presentation in places. For example, it is not clear to me why N2A cells were used to screen the capacity of different PUFAs to increase PIEZO2 rather than just complete these experiments in DRG neurons from AS mice which is the more directly relevant preparation?

3. Figure 9 cartoon. LA significantly increases PIEZO2 currents in WT neurons. I think this should be reflected in the cartoon as it suggests that in WT cells LA either results in an increase in the fraction of activatable channels or increases the channel open probability.

4. Please provide loading controls for the Western blots in the figures.

5. Please check the bar chart in Fig. 3e for a possible formatting error.

Reviewer #3 (Remarks to the Author):

Angelman syndrome (AS) is a neurodevelopmental disorder caused by loss of maternal UBE3A that leads to a number of impairments, including motor impairments. Characterizing causal factors in motor impairment and developing therapeutics to improve motor function have important clinical benefits for both researchers and patients. Romero et al. show that mechanosensitive PIEZO2-mediated activity is decreased in DRG neurons in AS mutant mice. They demonstrate in mouse DRG neurons and in AS patient cells that linoleic acid (LA) improves PIEZO2-mediated electrophysiological phenotypes. In AS model mice, they also provide evidence suggesting that a diet high in LA improves both PIEZO2-mediated currents and abnormal gait. Effects of LA on PIEZO2 do not appear to be specific to AS model mice, but PIEZO2 may be considered as potential therapeutic target regardless. The manuscript also provides evidence that loss of UBE3A affects PIEZO2 currents through a mechanism involving cofilin and actin. The link between this mechanism and gait impairments is not tested. Experiments across three models: Ube3am-/p+ mice, human cells with siRNA-mediated UBE3A knockdown, and stem cell-derived neurons from patients with AS is impressive and strengthens the impact of the work significantly. Overall, this work represents a timely and impactful contribution to understanding of motor impairments in Angelman syndrome, and identifies a novel potential therapeutic target. We have some comments and concerns about data collection and interpretation in certain areas:

Major comments:

1. The data in Figure 1 suggest that mechanosensitive PIEZO2 currents are impaired in Ube3am-/p+ mouse DRG neurons. Two pieces of supporting evidence would help to support this important result. First, are electrophysiological changes specific to mechanocurrents, or do they also generalize to intrinsic excitability (via current injection)? Figures S1h-k address this key point, but with a sample size that appears to be underpowered (n=5-8) relative to the sample sizes used for mechanosensitive currents (n=9-15). Therefore, it is difficult to evaluate the negative results. Second, are PIEZO2 levels decreased in Ube3a mutants? Western blotting of PIEZO2 as the authors have done in other places would directly address this important question.

2. The breeding scheme used to generate experimental animals for behavioral studies in Figures 6-7 is not clear based on the details provided in Methods. Were littermate controls used, and were experimenters blind to genotype? Since there is no WT + HF diet group, how is this possible with littermates/blinding? The rotarod also only includes no WT + LA diet group.

3. The data demonstrate that LA enhances PIEZO2 activity (Fig. 4), but does not increase the expression of PIEZO2 in the membrane (Fig. S4). It seems as though the mechanism by which LA regulates PIEZO2 currents ("decreases membrane structural order") is important enough to feature in the main figures. Figs S4f-j and S5 can be combined into a single main figure to highlight this mechanism for readers.

4. The model in Figure 9 attempts to put together many findings into a model about the mechanism by which LA improves PIEZO2 currents in Ube3a mutant mice. (Line 347: "We propose a mechanism whereby loss of UBE3A expression leads to an increase in cofilin, which severs actin filaments, and in turn, decreases PIEZO2 membrane expression and currents in the context of AS (Fig. 9).") This model is nice for readers though it does imply certain things that have not been explicitly tested (e.g. in vivo, the mechanism of action of LA on behavior was not shown to be independent of actin/cofilin). The Discussion would benefit from a paragraph acknowledging the gaps between what was demonstrated experimentally in this study and what is shown in this model, and what type of future work may address these gaps.

Minor comments:

Please include appropriate statistics in the results section or figure legends (F or t, p, and main effects of ANOVAs). Readers are unlikely to click through to the raw data table and this information should be accessible.

In the siRNA experiments, when measuring increases or decreases in protein expression, often claims are made about changes in expression of cofilin, actin, PIEZO2. These changes look meaningful, but no statistical comparisons are included (fold change relative to 1). (Figures 2b, 2d, 2f, 2g, 2h; Supplementary Figures 2d, 3c, 4f).

P2, L37: report human cell type and method used knock-down UBE3A expression in human cells in abstract.

Provide justification for the dose of LA that was used in mice and if it can be compared to doses administered on human cell lines.

In Figure 8h, what is the effect of LA treatment on neurotypical cells?

The discussion should also include more on the limitations of the LA fatty acid supplement in vivo, especially with regard to the high doses necessary to equal uptake as observed in vitro.

In the model in Fig 9, the mechano AP in AS mice is not smaller as pictured, but rather takes more stimulation to elicit. Is there a better way to illustrate this accurately in the model?

RESPONSE TO REVIEWER COMMENTS

We thank the editor and reviewers for their time invested in our manuscript and their constructive comments. Our work has improved thanks to their critiques, questions, and suggestions. Please find below a point-by-point response to the specific issues raised by the reviewers. Based on the reviewers' comments, we have added new information to the manuscript (highlighted in yellow).

Reviewer #1

(Remarks to the Author): Romero et al studies the role of Piezo2 channel in the development of ataxia in Angelman syndrome. Angelman syndrome is caused by loss of function of maternally derived UBE3A, a ubiquitin ligase. Angelman syndrome causes delayed development, intellectual disability as well as ataxia. The ataxic phenotype is attributed to cerebellar disorder, but there are data indicating that cerebellar dysfunction is only partially responsible for ataxic gait. Piezo2 is a mechanically activated ion channel, and the major symptom of its loss of function mutations in humans is ataxia.

The manuscript presents three major findings. 1. The authors show that in a mouse model of Angelman syndrome (Ube3a maternal KO), as well as in stem cell-derived neurons from Angelman syndrome patients, mechanically activated currents in DRG neurons are decreased. Ube3a knockdown in a Merkel cell carcinoma line also decreased Piezo2 mediated currents. 2. Linoleic supplementation increased Piezo2 mediated currents in Ube3A deficient (and wild type) cells, and alleviated the gait problems (increased stride length) in a mouse model of Angelman syndrome. 3. The authors propose a mechanism in which lack of Ube3a causes reduced ubiquitination of cofilin, leading to decreased actin levels which results in reduced Piezo2 currents.

I think the manuscript is important, as it addresses the mechanism of some of the symptoms of a devastating disease. The authors propose a feasible pathomechanism, as well as a potential therapy that is potentially very simple to implement. The data are solid, rigorous and convincing, and I strongly support publication after addressing the comments below.

1. Neither the effect of Ube3a reduction nor the effects linoleic acid are specific to Piezo2 or to rapidly adapting mechanically activated currents in DRG neurons. The logical link between Piezo2 and the ataxic phenotype of Angelman syndrome is provided by the knowledge that Piezo2 loss of function mutations in humans cause ataxia. a) Do children with Angelman syndrome have reduced light touch? If so, please discuss. Reduced light touch is also a symptom of Piezo2 loss of function mutations in humans. Also, b) do Ube3a knockout mice have other signs of Piezo2 deficiency at the behavioral level, such as reduced sensitivity in the von Frey assay?

The reviewer raises a good point. Unfortunately, there is a limited number of published studies quantifying the sensory behavior of individuals with Angelman syndrome (AS). A study reporting sensory processing abnormalities in persons with AS revealed that they tend to be hypo-responsive to tactile and vestibular/proprioceptive input (e.g., delayed responses to pain and put objects or toys in their mouth)¹. However, the authors did not explicitly report responses to touch but rather behaviors that could result from various neurobiological defects. Hence, it is unclear whether AS individuals display reduced touch sensitivity.

A previous work demonstrated that wild-type (WT) and AS mice responded similarly to von Frey filaments². Likewise, we found that WT, *Ube3a*^{m-/p+}, and *Ube3a*^{m+/p-} respond similarly to von Frey filaments (data not shown). These results support the idea that the reduced levels of PIEZO2 function in AS are sufficient to evoke normal responses to light touch, unlike PIEZO2 conditional knockout mice and humans containing a nonfunctional PIEZO2 variant or lacking PIEZO2 mRNA^{3,4}.

2. Figure 2 SiRNA experiments. The effect of siRNA on the Western is quite small, the effect on currents is a lot clearer.

a) Was there a fluorescent marker for transfection used? If so, please state, as it can explain the difference between the effects size on the currents and the Western blot, assuming that only fluorescent cells were patched, and transfection efficiency was not 100%.

We used siGLO Green as a transfection marker for patch-clamp experiments and recorded from green cells; however, we did not sort cells for protein expression experiments. As the reviewer pointed out, this could explain the observed difference between current and membrane protein expression reductions. We have now added the following sentence in the Results section (lines 139-143): “For electrophysiology experiments, we patched cells expressing the transfection marker siGLO green, whereas for western blots, we extracted membrane protein from a mixed culture of transfected and untransfected cells. This could explain the difference in effect size between currents and membrane protein expression reduction (47% vs. 23%).”

b) The gel pictures shows suggest a more than 50% reduction in intensity, but the summary figure shows ~50 % as the largest reduction. I would double check the numbers.

We thank the reviewer for noticing this oversight. We have now included a representative western blot that better reflects the quantification (Figure 2b top, and Supplementary Figure 2e).

c) Also, the significance between control and siRNA treatment is not noted. I think that is a more relevant comparison than the lack of significant difference between Piezo2 siRNA and Ube3a siRNA. The same applies for panels f,g,h, where fold changes are shown, significance should be show, one sample t-test, difference from 1, or paired t-test between individual values would suffice, provided that the data are normally distributed.

We thank the reviewer for this suggestion, we are now comparing the differences between control and treatments, using one sample t-test, for all the western blot results in the manuscript.

Minor comments:

3. I would show exact p values including with those labeled ns. In most cases the effects are very clear, but in some panels such as Figure 4 b,c there are some visible trends in some of the “ns” panel, and the actual effect is only one asterisk, so exact p values would be informative.

In the revised version of the manuscript, all the *p*-values are shown on each panel instead of asterisks or n.s.

4. Some of the statistical comparisons were tested with parametric test. i.e. t-test and ANOVA, some with non-parametric. Please explain if this was based of testing normality of data distribution.

We used parametric and non-parametric tests according to the data distribution. In the revised method’s section, we now included the following statement: “We used the Kolmogorov and Smirnov method to determine data distribution, as well as the Bartlett’s test to determine differences between standard deviations.” (lines 808-810).

5. Title: I think Angelman syndrome denotes the human condition. I would change the title to “...mouse model of Angelman Syndrome”, or similar. Also the title describe a subset of findings, the authors may consider a title that is more inclusive of the rest of the findings.

We have changed the title to “Linoleic Acid Improves PIEZO2 Dysfunction in a Mouse Model of Angelman Syndrome”. In our opinion, this title combines the two major discoveries in the manuscript, that PIEZO2 function is reduced in AS and that linoleic acid enhances its activity.

6. Abstract, I would clarify that Piezo2 is a mechanically activated ion channel.

The abstract now reads in the third sentence: “PIEZO2 is a mechanosensitive ion channel essential for coordination and balance”

Reviewer #2 (Remarks to the Author):

Angelman syndrome (AS) is a neurogenetic disorder characterized by intellectual disability and movement deficits that arises as a consequence of loss of expression of UBE3 E3 ligase from the maternal allele. Given that both individuals with AS and those with loss of function of PIEZO2 display impaired balance, coordination and gait, the authors here set out to determine whether sensory neuron PIEZO2 is attenuated in AS and conditions in which UBE3 expression is suppressed. They found that PIEZO2 currents are significantly reduced in sensory neurons from a mouse model of AS, in cells in which UBE3 is knocked down, as well as human stem-cell derived neurons from AS patients. They link the decrease in PIEZO2 to a decrease in F-actin that is downstream of enhanced expression of cofilin, an actin binding protein that promotes actin disassembly. They further found that the deficits in PIEZO2 in neurons from AS mice or derived from human stem cells can be reversed by incubation with linoleic acid (LA) which acts by modifying the membrane properties in a manner that is sensed by PIEZO2 but not other membrane channels. Finally, they showed that AS mice fed with LA-enriched diet displayed normalized PIEZO2 currents and improved gait.

Overall, this study is impressive in its scope and translational potential. The experiments are technically well done and much of the interpretation is supported by the data. There are some issues that need to be addressed to strengthen some of the conclusions.

1. One area in which the manuscript is unclear is in the mechanism of action of linoleic acid in selectively enhancing PIEZO2 currents. The authors suggest that this is due to membrane remodeling, which is a rather non-specific effect that might be expected to alter the properties of multiple membrane ion channels which is not observed. Do none of the other PUFAs tested in Fig. 4 (which are not effective in increasing PIEZO2 currents) change membrane properties? Unfortunately, the DSC experiments conducted for LA (Supplemental Figs. 4i and 4j) were not done for these other PUFAs to address this question.

We would like to clarify that, in the manuscript, we did not claim that LA is selectively enhancing PIEZO2. Indeed, we showed that LA also enhances the function of the intermediate and slow inactivating mechanocurrents (whose protein identities have not yet been determined). Moreover, we also show that the function of a bacterial mechanosensitive ion channel (MscL), whose activation solely relies on the membrane's mechanical properties, can be increased using LA.

The exquisite sensitivity displayed by mechanosensitive ion channels to small changes in the mechanical properties of the plasma membrane is unlikely shared by other membrane proteins (*at least to the same extent*). Even within mechanosensitive responses, we determined that the LA-enriched diet did not increase slowly inactivating currents (Fig. 7c, right panel). However, further supplementing DRG neurons from *Ube3a^{m-/p+}* mice fed with a LA-enriched diet with additional LA (during culture) increased the slowly inactivating currents (Supplementary Fig. 6c-d). Although we demonstrated a lack of effect of LA on some TRP channels and current elicited action potentials (Supplementary Fig. 5), future studies could focus on evaluating other membrane proteins and ion channels (e.g., GPCRs).

We thank the reviewer for this suggestion. We have added new DSC data (Fig. 5e-h) with the other fatty acids used for the electrophysiology recordings. Moreover, we also included new analyses highlighting LA's differential effect on membrane mechanical properties compared to the other fatty acids (lines 216-231).

2. The study utilizes a variety of cell types – DRG neurons, MCC cells, N2A cells, stem-cell-derived neurons. On the one hand, it is impressive and reinforcing that the correlation between UBE3 expression and PIEZO2 current amplitude persists across all these different cell types. On the other hand, I don't think the use of all of them is sufficiently justified in the manuscript and, in my view, they detract from the overall presentation in places. For example, it is not clear to me why N2A cells were used to screen the capacity of different PUFAs to increase PIEZO2 rather than just complete these experiments in DRG neurons from AS mice which is the more directly relevant preparation?

As noted by the reviewer, we tested our hypothesis in various cell lines to demonstrate that our results were not dependent on the chosen cell system but rather dependent on the relationship between UBE3A and the function of PIEZO2.

We understand the reviewer's comment. Here are the practical reasons to justify our choice of cell types:

- 1- Mouse dorsal root ganglia (DRG) and human stem cell-derived neurons (i.e., from neurotypical and individuals with AS) were used to evaluate the effect of loss of expression of *Ube3a* on mechanosensation in mouse and human models of AS.
- 2- We used human MCC13 cells because all mechanocurrents in these cells are encoded by *Piezo2*⁵. Unfortunately, antibodies against PIEZO2 have limited efficacy in recognizing PIEZO2 in the membrane fraction of mouse DRG neurons^{6,7}. On the other hand, human MCC13 cells allowed us to perform biochemical experiments. In the new version of the manuscript, we added the following sentence, "To support functional and biochemical experiments, we utilized the human Merkel cell carcinoma cell line (MCC13), in which PIEZO2 mediates all endogenous mechanosensitive currents" (lines 132-137).
- 3- *Piezo2*-transfected N2A cells (i.e., *Piezo1*^{-/-} N2A cells) only display the rapidly inactivating currents. This feature allows us to unequivocally distinguish the effect of various fatty acids on PIEZO2 activation and inactivation rather than on other mechanocurrents (i.e., intermediate and slow inactivating currents). In the new version of the manuscript, we added the following sentence, "To this end, we transfected *Piezo2* variant 14 (abundant in the mouse trigeminal ganglion)⁸ into N2A cells lacking *Piezo1* (*Piezo1*^{-/-} N2A cells)⁹ to distinguish the effect of LA on PIEZO2 gating unequivocally." (lines 193-195).
- 4- We used iPSC-derived sensory neurons to validate our hypothesis in human sensory neurons expressing PIEZO2 endogenously.

3. Figure 9 cartoon. LA significantly increases PIEZO2 currents in WT neurons. I think this should be reflected in the cartoon as it suggests that in WT cells LA either results in an increase in the fraction of activatable channels or increases the channel open probability.

We would like to maintain the current cartoon version since it highlights the major discoveries. Although we agree with the reviewer that LA increases mechanocurrents, it did not change the mechano-action potentials or gait behavior in the WT. For these reasons, we prefer to keep the three panels to avoid diluting the main message.

4. Please provide loading controls for the Western blots in the figures.

For western blot analysis, we used Image Lab Software (Bio-Rad) to measure each chemiluminescent signal vs. the total protein loaded (stain-free signal, shown in the supplementary figures) in the corresponding lane and then normalized with the control treatment^{10,11}. We prefer to use total loaded protein per well to quantify the band of interest and compare the intensities between conditions. Many journals favor this method rather than the classical normalization with housekeeping proteins. We feel this is particularly important because the loss of UBE3A could affect other proteins, as we show with F-actin and cofilin.

These details are now included in the Methods section: "Western blots were analyzed, using Image Lab Software (Bio-Rad), by measuring each chemiluminescent signal vs. the total protein loaded (stain-free signal) in the corresponding lane^{10,11}. We reported band relative intensities, and for experiments utilizing MCC13 cells, we normalized against the control treatment." (lines 629-632).

5. Please check the bar chart in Fig. 3e for a possible formatting error.

Thank you for noticing this mishap. We have fixed the panel in the new version.

Reviewer #3 (Remarks to the Author):

Angelman syndrome (AS) is a neurodevelopmental disorder caused by loss of maternal UBE3A that leads to a number of impairments, including motor impairments. Characterizing causal factors in motor impairment and developing therapeutics to improve motor function have important clinical benefits for both researchers and patients. Romero et al. show that mechanosensitive PIEZO2-mediated activity is decreased in DRG neurons in AS mutant mice. They demonstrate in mouse DRG neurons and in AS patient cells that linoleic acid (LA) improves PIEZO2-mediated electrophysiological phenotypes. In AS model mice, they also provide evidence suggesting that a diet high in LA improves both PIEZO2-mediated currents and abnormal gait. Effects of LA on PIEZO2 do not appear to be specific to AS model mice, but PIEZO2 may be considered as potential therapeutic target regardless. The manuscript also provides evidence that loss of UBE3A affects PIEZO2 currents through a mechanism involving cofilin and actin. The link between this mechanism and gait impairments is not tested. Experiments across three models: Ube3am-/p+ mice, human cells with siRNA-mediated UBE3A knockdown, and stem cell-derived neurons from patients with AS is impressive and strengthens the impact of the work significantly. Overall, this work represents a timely and impactful contribution to understanding of motor impairments in Angelman syndrome, and identifies a novel potential therapeutic target. We have some comments and concerns about data collection and interpretation in certain areas:

Major comments:

1. The data in Figure 1 suggest that mechanosensitive PIEZO2 currents are impaired in Ube3am-/p+ mouse DRG neurons. Two pieces of supporting evidence would help to support this important result.

First, are electrophysiological changes specific to mechanocurrents, or do they also generalize to intrinsic excitability (via current injection)? Figures S1h-k address this key point, but with a sample size that appears to be underpowered (n=5-8) relative to the sample sizes used for mechanosensitive currents (n=9-15). Therefore, it is difficult to evaluate the negative results.

We agree with this reviewer's comment. In the new version of the manuscript, we have increased the number of samples (n= 15) to further support the notion that current-elicited action potentials are not affected in the DRG neurons from *Ube3a^{m-/p+}* mice (Supplementary Fig. 1h-k).

Second, are PIEZO2 levels decreased in Ube3a mutants? Western blotting of PIEZO2 as the authors have done in other places would directly address this important question.

We show that PIEZO2 membrane expression decreases after knocking down *UBE3A* in human MCC13 cells using an antibody against human PIEZO2. Unfortunately, antibodies against PIEZO2 have limited efficacy for recognizing PIEZO2 in the membrane fraction of mouse DRG neurons^{6,7}. This impairs our ability to determine biochemically PIEZO2 membrane content in mouse DRG neurons. The lack of mouse PIEZO2 antibodies is a known issue in the field, and we wish we could answer this question more definitively.

2. The breeding scheme used to generate experimental animals for behavioral studies in Figures 6-7 is not clear based on the details provided in Methods.

While the majority of mice used for these studies were purchased from the breeding colony maintained at JAX laboratories for the *Ube3a^{tm1Alb}* mutant, we also bred mice in-house for our studies. As stated now in the methods: "Male and female heterozygous *Ube3a^{m-/p+}* maternal- and *Ube3a^{m+/p-}* paternal-transmission knockout mice were obtained from The Jackson Laboratory, strains C57BL/6 *Ube3a^{tm1Alb}* (B6 AS; Stock No. 016590). C57BL/6J mice (WT) were obtained from The Jackson Laboratory (Stock No. 000664). We also bred mice from our in-house colony at UTHSC and their genotype was confirmed using previously published protocols^{12, 13}. We used a two-step breeding scheme. Briefly, we first crossed female WT with male *Ube3a^{m+/p-}* to generate heterozygous offspring, in which half of the progeny were *Ube3a^{m+/p-}*, with no AS phenotype. *Ube3a^{m+/p-}* heterozygous females were crossed with WT males to generate *Ube3a^{m-/p+}* mice, in which half of the progeny

constitute AS mice displaying the corresponding phenotype. After confirming the genotype, the progenies were used for experiments.” (lines 451-460)

Were littermate controls used, and were experimenters blind to genotype? Since there is no WT + HF diet group, how is this possible with littermates/blinding?

The *Ube3a* gene is imprinted in mammals, requiring separate crosses to generate *Ube3a^{m-/p+}* vs *Ube3a^{m+/p-}* animals (as described above). We used both *Ube3a^{m-/p+}* and *Ube3a^{m+/p-}* animals on the same strain background generated at the same facility. Since we wanted to include the paternal inheritance of the *Ube3a* allele, also, as a control for behavioral experiments, it was not feasible to test animals from the same cross. Although many studies in the field use an F1 mixed BL6 control, we would argue that using animals in the same genetic background would be appropriate while only changing the inherited allele (maternal vs. paternal deficiency). We have checked our gait data against a recent publication, using the F1 controls, that showed gait differences in a mouse model of AS¹⁴. Petkova *et al.* reported a difference of ~13% between the stride length of F1 controls and *Ube3a^{m-/p+}* littermate mice, which is within the range of the difference we observed of ~12-15% in our behavioral work using non-littermate mice.

In addition, we could not breed littermates that included F1 controls, WT, *Ube3a^{m-/p+}*, and *Ube3a^{m+/p-}* mice on three different diets (≥150 animals). Purchasing some of these animals was our only option to carry out the patch-clamp electrophysiology (voltage- and current-clamp), imaging, mass spectrometry, western blots, and behavioral experiments. Noteworthy, the electrophysiology data shown in Supplementary Fig. 1f-g were obtained from WT and *Ube3a^{m-/p+}* littermate mice; the differences in mechanocurrents and thresholds recapitulate the differences observed in non-littermate mice. Finally, we show that *Ube3a^{m-/p+}* animals alone respond to a dietary intervention (e.g., same mice with the same genetic background + or – dietary intervention).

For electrophysiology, the investigator was blind to genotype and treatment. For behavioral assays, the investigators were blind when possible; however, diet smells, and consistencies can be easily identified. Moreover, the fur of mice on oily-based diets (i.e., linoleic acid) looks shinier as they get covered with the semisolid diet when compared to the other diets. However, gait dynamics were quantified using the CatWalkXT (Noldus, USA) instrument. The catwalk analysis requires minimal human intervention. The main intervention is moving the animals from their cages to the catwalk scanner. Each mouse could walk freely in both directions, and the instrument records and provides the values, which were later plotted. In our opinion, this type of behavior and data acquisition reduces the implicit bias of the experimentalist to a minimum.

The rotarod also only includes no WT + LA diet group.

The purpose of supplementary figure Fig 8e is to show that LA treatment had no effect on rotarod performance for *Ube3a^{m-/p+}* animals. The WT control is an untreated reference for rotarod performance. The rotarod experiments were not done blind to genotype.

4. The model in Figure 9 attempts to put together many findings into a model about the mechanism by which LA improves PIEZO2 currents in Ube3a mutant mice. (Line 347: "We propose a mechanism whereby loss of UBE3A expression leads to an increase in cofilin, which severs actin filaments, and in turn, decreases PIEZO2 membrane expression and currents in the context of AS (Fig. 9).") This model is nice for readers though it does imply certain things that have not been explicitly tested (e.g. in vivo, the mechanism of action of LA on behavior was not shown to be independent of actin/cofilin). The Discussion would benefit from a paragraph acknowledging the gaps between what was demonstrated experimentally in this study and what is shown in this model, and what type of future work may address these gaps.

We have now added the limitations of our study in the discussion section (lines 429-437), as follows:

“In this work, we focused on determining the mechanism whereby UBE3A regulates mechanosensation in AS. Even though we provided evidence that loss of *UBE3A* expression decreases PIEZO2 membrane expression and currents, we do not have direct evidence demonstrating that the reduction of PIEZO2 function is the main cause of the gait phenotype in AS. More experimental insight will be needed to determine the full implications of the reduction of PIEZO2 on gait in the context of AS. Although we found that a LA-enriched diet increases PIEZO2 activity and mechano-excitability, as well as improves gait in AS mice, our work does not rule out that the effect of LA on behavior could occur somewhere else beyond solely increasing PIEZO2 function in DRG neurons. Future experiments could be directed to test the effect of the LA diet on neuronal *Piezo2* conditional knockout mice.”

Minor comments:

Please include appropriate statistics in the results section or figure legends (F or t, p, and main effects of ANOVAs). Readers are unlikely to click through to the raw data table and this information should be accessible.

We have included the statistical details in the figure panels and legends in the new manuscript version.

In the siRNA experiments, when measuring increases or decreases in protein expression, often claims are made about changes in expression of cofilin, actin, PIEZO2. These changes look meaningful, but no statistical comparisons are included (fold change relative to 1). (Figures 2b, 2d, 2f, 2g, 2h; Supplementary Figures 2d, 3c, 4f).

Based on these reviewers' suggestions, we are now comparing the differences between control and treatments for all the western blots in the manuscript.

P2, L37: report human cell type and method used knock-down UBE3A expression in human cells in abstract.

In the abstract, we have now added: "...cultured human cells with *UBE3A* knock-down...". Unfortunately, the 150-word limit makes it very difficult to add human Merkel cell carcinoma cell line (MCC13) and reprogrammed human-induced pluripotent stem cells (iPSCs) into sensory neurons. If the editor agrees, we will happily include this information in the abstract.

Provide justification for the dose of LA that was used in mice and if it can be compared to doses administered on human cell lines.

We designed a non-western-style (i.e., low sugars) high-fat diet enriched with 59% of fat-derived calories from safflower oil as a delivery method to maximize the LA content (now written in lines 377-380) based on the following information:

- 1) We received support from the company Dyets Inc. to design a diet containing high-fat levels similar to what has been used in the literature (including ketogenic diets).
- 2) A ketogenic dietary intervention has been shown to reduce seizures in individuals with AS^{15, 16, 17, 18}. Ketogenic diets deliver 80-90% of its calories from fat^{19, 20}. The ketogenic diet is a well-established non-pharmacological approach to treat drug-resistant epilepsy in children²¹.

We think that it is not possible to compare the dosage of LA administered in culture and *in vivo*. On the one hand, cultured cells are constantly exposed to the media enriched in LA, whereas *in vivo*, the LA is absorbed and delivered by the digestive and circulatory systems, respectively. In our opinion, a fair comparison between the amount of LA present in membranes after culture and diet supplementation can be made using liquid chromatography-mass spectrometry (LC-MS). Indeed, we determined that LA accumulates in membranes

sevenfold when supplemented in culture media versus twofold when fed to mice (Supplementary Fig. 4c and Fig. 7a).

In Figure 8h, what is the effect of LA treatment on neurotypical cells?

We did not measure the effect of LA on DPSC-derived neurons from neurotypical individuals. However, we expect LA to have the same enhancing effect shown for iPSC-derived neurons from neurotypical individuals, wild-type MCC13 cells, mouse DRG neurons, and N2A cells.

The discussion should also include more on the limitations of the LA fatty acid supplement in vivo, especially with regard to the high doses necessary to equal uptake as observed in vitro.

We did not design the diet strategy to equal the uptake observed *in vitro*. Instead, we used a non-western-style diet enriched in safflower oil as a delivery method to increase LA in mouse DRG neurons and to provide proof of concept that a LA-enriched diet could be used to modulate the function of sensory receptors. As mentioned above, we think it is not feasible to compare the dosage of LA administered in culture and *in vivo*. To ameliorate the reviewers' concerns, we have *deleted* the sentence of the discussion: “suggesting that a LA dietary intervention has the potential to improve gait in individuals with AS”.

Future experiments could be directed to test alternative delivery methods (e.g., diets with lower LA content and/or various feeding periods). In our opinion, these experiments are outside the scope of this initial communication.

In the model in Fig 9, the mechano AP in AS mice is not smaller as pictured, but rather takes more stimulation to elicit. Is there a better way to illustrate this accurately in the model?

We agree with the reviewer and have modified the model accordingly.

References:

1. Walz NC, Baranek GT. Sensory processing patterns in persons with Angelman syndrome. *Am J Occup Ther* **60**, 472-479 (2006).
2. McCoy ES, Taylor-Blake B, Aita M, Simon JM, Philpot BD, Zylka MJ. Enhanced Nociception in Angelman Syndrome Model Mice. *J Neurosci* **37**, 10230-10239 (2017).
3. Chesler AT, *et al.* The Role of PIEZO2 in Human Mechanosensation. *N Engl J Med* **375**, 1355-1364 (2016).
4. Ranade SS, *et al.* Piezo2 is the major transducer of mechanical forces for touch sensation in mice. *Nature* **516**, 121-125 (2014).
5. Shin KC, *et al.* The Piezo2 ion channel is mechanically activated by low-threshold positive pressure. *Sci Rep* **9**, 6446 (2019).
6. Shin SM, *et al.* Piezo2 mechanosensitive ion channel is located to sensory neurons and nonneuronal cells in rat peripheral sensory pathway: implications in pain. *Pain* **162**, 2750-2768 (2021).
7. Woo SH, *et al.* Piezo2 is the principal mechanotransduction channel for proprioception. *Nat Neurosci* **18**, 1756-1762 (2015).

8. Szczot M, *et al.* Cell-Type-Specific Splicing of Piezo2 Regulates Mechanotransduction. *Cell Rep* **21**, 2760-2771 (2017).
9. Moroni M, Servin-Vences MR, Fleischer R, Sanchez-Carranza O, Lewin GR. Voltage gating of mechanosensitive PIEZO channels. *Nat Commun* **9**, 1096 (2018).
10. Aldridge GM, Podrebarac DM, Greenough WT, Weiler IJ. The use of total protein stains as loading controls: an alternative to high-abundance single-protein controls in semi-quantitative immunoblotting. *J Neurosci Methods* **172**, 250-254 (2008).
11. Wahl D, *et al.* Comparing the Effects of Low-Protein and High-Carbohydrate Diets and Caloric Restriction on Brain Aging in Mice. *Cell Rep* **25**, 2234-2243 e2236 (2018).
12. Jiang YH, *et al.* Mutation of the Angelman ubiquitin ligase in mice causes increased cytoplasmic p53 and deficits of contextual learning and long-term potentiation. *Neuron* **21**, 799-811 (1998).
13. Judson MC, *et al.* GABAergic Neuron-Specific Loss of Ube3a Causes Angelman Syndrome-Like EEG Abnormalities and Enhances Seizure Susceptibility. *Neuron* **90**, 56-69 (2016).
14. Petkova SP, Adhikari A, Berg EL, Fenton TA, Duis J, Silverman JL. Gait as a quantitative translational outcome measure in Angelman syndrome. *Autism Res* **15**, 821-833 (2022).
15. Evangeliou A, Doulioglou V, Haidopoulou K, Aptouramani M, Spilioti M, Varlamis G. Ketogenic diet in a patient with Angelman syndrome. *Pediatr Int* **52**, 831-834 (2010).
16. Thibert RL, *et al.* Epilepsy in Angelman syndrome: a questionnaire-based assessment of the natural history and current treatment options. *Epilepsia* **50**, 2369-2376 (2009).
17. Valente KD, *et al.* Epilepsy in patients with angelman syndrome caused by deletion of the chromosome 15q11-13. *Arch Neurol* **63**, 122-128 (2006).
18. Thibert RL, *et al.* Low glycemic index treatment for seizures in Angelman syndrome. *Epilepsia* **53**, 1498-1502 (2012).
19. Weber DD, Aminzadeh-Gohari S, Tulipan J, Catalano L, Feichtinger RG, Kofler B. Ketogenic diet in the treatment of cancer - Where do we stand? *Mol Metab* **33**, 102-121 (2020).
20. Sethuraman A, Rao P, Pranay A, Xu K, LaManna JC, Puchowicz MA. Chronic Ketosis Modulates HIF1alpha-Mediated Inflammatory Response in Rat Brain. *Adv Exp Med Biol* **1269**, 3-7 (2021).
21. Gasior M, Rogawski MA, Hartman AL. Neuroprotective and disease-modifying effects of the ketogenic diet. *Behav Pharmacol* **17**, 431-439 (2006).

Reviewer #1 (Remarks to the Author):

The authors provided satisfactory responses to the critiques, I recommend acceptance.

Reviewer #2 (Remarks to the Author):

The authors have been very responsive to critiques. I have no further concerns on the manuscript. Congratulations to the authors for a very nice study.

Reviewer #3 (Remarks to the Author):

This revision thoughtfully addresses most of my initial comments and concerns.

Regarding major concern #2: The explanation of the limits of this breeding scheme make sense, and I understand why it is not possible to have WT, UBE3A m-/p+, and UBE3A m+/p- mutants all from the same litter. The authors argue in the rebuttal that the advantages of including the UBE3A m+/p- mice outweigh the disadvantages of not having littermate controls. The authors should add a sentence to the manuscript (either in the methods or discussion) acknowledging the advantages and disadvantages of this design. I also now understand that some experimental mice were ordered directly from Jackson Labs and others were bred in house. This information is mentioned broadly in the rebuttal, but not in detail in the manuscript. Please include details in the Methods section or in a supplementary table on exactly which experiments used Jax mice, which experiments used in house mice, which experiments used littermate controls, and which did not. This transparency is important for readers to evaluate results.

RESPONSE TO REVIEWER COMMENTS

Reviewer #1 (Remarks to the Author):

The authors provided satisfactory responses to the critiques, I recommend acceptance.

Reviewer #2 (Remarks to the Author):

The authors have been very responsive to critiques. I have no further concerns on the manuscript. Congratulations to the authors for a very nice study.

We thank you for your time invested in our manuscript and your constructive comments.

Reviewer #3 (Remarks to the Author):

This revision thoughtfully addresses most of my initial comments and concerns.

Regarding major concern #2:

The explanation of the limits of this breeding scheme make sense, and I understand why it is not possible to have WT, UBE3A m-/p+, and UBE3A m+/p- mutants all from the same litter. The authors argue in the rebuttal that the advantages of including the UBE3A m+/p- mice outweigh the disadvantages of not having littermate controls. The authors should add a sentence to the manuscript (either in the methods or discussion) acknowledging the advantages and disadvantages of this design.

In the methods section, we have now included a paragraph named *control animals*. This paragraph highlights the rationale behind using paternal mice. We have also added, in the result section, the comparison between our non-littermate results with the recent publication from Petkova et al. with WT and *Ube3a*^{m-/p+} littermates.

I also now understand that some experimental mice were ordered directly from Jackson Labs and others were bred in house. This information is mentioned broadly in the rebuttal, but not in detail in the manuscript. Please include details in the Methods section or in a supplementary table on exactly which experiments used Jax mice, which experiments used in house mice, which experiments used littermate controls, and which did not. This transparency is important for readers to evaluate results.

Based on this reviewer's suggestion, we have now included (in the methods section, *control animals*) the details regarding which experiments used JAX mice, in house, or littermates.